# Single-Base Methylome Analysis of Sweet Cherry (*Prunus avium* L.) on Dwarfing Rootstocks Reveals Epigenomic Differences Associated with Scion Dwarfing Conferred by Grafting

**DOI:** 10.3390/ijms252011100

**Published:** 2024-10-16

**Authors:** Yi Hong, Zhuang Wen, Guang Qiao, Tian Tian, Xiaopeng Wen

**Affiliations:** Key Laboratory of Plant Resource Conservation and Germplasm Innovation in Mountainous Region (Ministry of Education), Institute of Agro-Bioengineering/College of Life Sciences, Guizhou University, Guiyang 550025, China; yhong1@gzu.edu.cn (Y.H.); gzu_zwen@163.com (Z.W.); 13518504594@163.com (G.Q.); tiantiangzu@163.com (T.T.)

**Keywords:** *Prunus tomentosa*, DNA methylation, whole-genome bisulfite sequencing, transcriptome, rootstock–scion interactions

## Abstract

Plant grafting using dwarfing rootstocks is one of the important cultivation measures in the sweet cherry (*Prunus avium*) industry. In this work, we aimed to explore the effects of the dwarfing rootstock “Pd1” (*Prunus tomentosa*) on sweet cherry ‘Shuguang2’ scions by performing morphological observations using the paraffin slice technique, detecting GA (gibberellin) and IAA (auxin) contents using UPLC-QTRAP-MS (ultra-performance liquid chromatography coupled with a hybrid triple quadrupole-linear ion trap mass spectrometer), and implementing integration analyses of the epigenome and transcriptome using whole-genome bisulfite sequencing and transcriptome sequencing. Anatomical analysis indicated that the cell division ability of the SAM (shoot apical meristem) in dwarfing plants was reduced. Pd1 rootstock significantly decreased the levels of GAs and IAA in sweet cherry scions. Methylome analysis showed that the sweet cherry genome presented 15.2~18.6%, 59.88~61.55%, 28.09~33.78%, and 2.99~5.28% methylation at total C, CG, CHG, and CHH sites, respectively. Shoot tips from dwarfing plants exhibited a hypermethylated pattern mostly due to increased CHH methylation, while leaves exhibited a hypomethylated pattern. According to GO (Gene Ontology) and KEGG (Kyoto Encyclopedia of Genes and Genomes) analysis, DMGs (differentially methylated genes) and DEGs (differentially expressed genes) were enriched in hormone-related GO terms and KEGG pathways. Global correlation analysis between methylation and transcription revealed that mCpG in the gene body region enhanced gene expression and mCHH in the region near the TSS (transcription start site) was positively correlated with gene expression. Next, we found some hormone-related genes and TFs with significant changes in methylation and transcription, including *SAURs*, *ARF*, *GA2ox*, *ABS1*, *bZIP*, *MYB*, and *NAC*. This study presents a methylome map of the sweet cherry genome, revealed widespread DNA methylation alterations in scions caused by dwarfing rootstock, and obtained abundant genes with methylation and transcription alterations that are potentially involved in rootstock-induced growth changes in sweet cherry scions. Our findings can lay a good basis for further epigenetic studies on sweet cherry dwarfing and provide valuable new insight into understanding rootstock–scion interactions.

## 1. Introduction

Sweet cherry (*Prunus avium* L.) belongs to the genus *Prunus* (Rosaceae) and is cultivated commercially in temperate regions in more than 40 countries around the world [1,2]. Sweet cherry has become one of the most popular fruits because of its delicious taste and high nutritional value [3]. Grafting is an ancient asexual propagation technique that most commonly involves joining two segments: the shoot segment is known as the “scion”, and the root segment is called the “rootstock” [4]. The rootstock affects a variety of physiological and biochemical characteristics of the scion, including growth vigor, fruit quality, fruit yield, and tolerance to various stresses [5]. Plant grafting using dwarfing rootstocks can achieve higher planting density, increase yield, facilitate pruning and harvesting, and allow more efficient application of pesticides, which is one of the important cultivation measures in the sweet cherry industry.

*Prunus tomentosa* Thunb., commonly known as Tomentosa cherry, is a Rosaceae plant adapted to diverse environments, which is native to temperate regions of China and has abundant germplasm resources in China [6]. *P. tomentosa* has been widely used as rootstock for sweet cherries. In our previous work, we screened and obtained a dwarf rootstock germplasm “Pd1” from the diverse germplasm resources of *P. tomentosa* in Guizhou Province, China. Compared with wild-type *P. tomentosa* rootstock, Pd1 exhibited an excellent dwarfing effect on sweet cherry scions. The application of grafting techniques using dwarfing rootstocks has a long history. Various hypotheses have explained the differences in scion vigor caused by dwarfing rootstocks, including that rootstocks can regulate the growth of scions by affecting the water and solute supply, hormone synthesis and transport, and anatomy of the graft union [7,8,9,10]. However, the molecular mechanism of the dwarfing rootstock’s influence on scion vigor is still unclear.

DNA methylation is a vital epigenetic modification and influences a variety of physiological processes in plants through the regulation of gene expression [11,12]. Currently, an increasing number of studies have demonstrated that DNA methylation is involved in rootstock–scion interactions and that grafting causes alterations in scion DNA methylation [13]. According to past research, global DNA methylation was significantly increased in cucumber and melon scions grafted onto pumpkin rootstocks [14]. In interspecific grafting of Solanaceae plants, global DNA methylation was altered in tomato, pepper, and eggplant scions [15]. Xanthopoulou et al. used different pumpkin cultivars for grafting and found that the rootstock affected the size of the scion fruit, accompanied by significant changes in scion DNA methylation [16]. In addition, pumpkin rootstock affected the methylation level of the transcription factor *CsWIN1* in cucumber scions, thereby regulating the biosynthesis of cucumber pericarp wax [17]. Uthup et al. revealed that rubber trees grafted onto rootstocks with different genetic backgrounds showed significant changes in DNA methylation levels [18]. Although these investigations provide some clues implicating DNA methylation in the grafting process [13,14,15,16,17,18], the mechanisms underlying DNA methylation-mediated changes in scion vigor induced by dwarfing rootstocks remain largely unknown.

In the present study, we aimed to explore the potential epigenetic regulation of Pd1 dwarfing rootstock’s influence on sweet cherry scion vigor. We observed paraffin sections of shoot tips, measured the growth vigor of new branches, detected the contents of gibberellins (GAs) and auxin (IAA) in shoot tips, generated single-base-resolution cytosine methylation maps of the genome and transcriptome sequencing atlases, and performed a comprehensive analysis of DNA methylation and the transcriptome. Our study was designed to gain insight into three questions: (i) the genomic landscape of the sweet cherry methylome, (ii) changes in the scion methylome associated with dwarfing rootstock, and (iii) evaluation of the relationship between methylome changes and gene expression changes of sweet cherry scions.

## 2. Results

### 2.1. Anatomical Structure of Shoot Tips in the Growth Stagnation Stage

The shoot apical meristem (SAM) determines the number of cells in the longitudinal direction, affecting branch elongation. Since the arrest of apical growth in SG-Pd1 (“Shuguang2” sweet cherry scions grafted onto Pd1 rootstocks) occurred at 195 days after grafting (Figure 1A), we used the paraffin sectioning technique to examine the shoot tips of SG-WT (‘Shuguang2” sweet cherry grafted onto wild-type rootstock) and SG-Pd1. Samples from SG-WT were used as controls. As shown in Figure 1B, vertical sections showed that the SAM of SG-Pd1 was bigger than that of SG-WT and was surrounded by lignified scales. Additionally, compared with the SG-WT, the cells in the SAM of SG-Pd1 were smaller, with lignified cell walls, and the cells in the rib region were more closely arranged, with smaller intercellular spaces. These findings indicated that the cell division ability of the SAM in dwarfing plants was reduced, resulting in restricted stem growth.

### 2.2. Effects of Dwarfing Rootstock on New Branch Growth

As shown in Figure 2A, after 195 days of grafting, the length and diameter of new branches in SG-Pd1 (dwarf) were significantly less than those in SG-WT (vigorous). In addition, there was no significant difference in the length and diameter of new branches from SG-Pd1 between 195 and 210 days, indicating that branches from SG-Pd1 stopped growing. Additionally, we measured new branch lengths, diameters, and internode lengths in four-year-old plants. As shown in Figure 2B, compared to the branches from SG-WT, the terminal and non-terminal branches from SG-Pd1 revealed shorter lengths, smaller diameters, and shorter internode lengths. These results indicated that Pd1 dwarfing rootstocks significantly reduced the growth vigor of “Shuguang2” sweet cherry.

### 2.3. The Content of Gibberellins and Auxin in Shoot Tips

Hormones play a crucial role in plant growth and development, with gibberellins and auxins being particularly important in regulating plant height. Our above findings showed that Pd1 dwarfing rootstocks affected the growth vigor of sweet cherry scions. Hence, we measured the content of gibberellins (GAs) and auxin (IAA) in the shoot tips. As shown in Figure 3, most types of GAs were observably decreased in their abundance in the shoot tips of SG-Pd1, and IAA was also dramatically decreased. These data indicated that Pd1 rootstock significantly reduced the levels of GAs and IAA in sweet cherry scions, which may be a primary cause of dwarfing.

### 2.4. The Expression of Genes Encoding DNA Methylation-Related Enzymes

DNA methylation-related enzymes are involved in the establishment, maintenance, and removal of methylation marks. Hence, we globally inspected the genes encoding methylation-related enzymes at the genome scale and analyzed the expression levels using RT-qPCR (real-time quantitative reverse transcription–polymerase chain reaction). Two *DRM2* and five *AGO4* genes responsible for de novo DNA methylation were found. One *MET1* gene, three *CMT3* genes, and other genes (*DDM1*, *SUVH4*, *SUVH5*, and *SUVH6*) involved in the maintenance of methylation were identified. In addition, one *ROS1* and five *DME* genes that are involved in demethylation were found. Next, the expression patterns of these genes were further analyzed using RT-qPCR in shoot tips (Figure 4). Compared with SG-WT, most genes showed reduced expression in SG-Pd1. Three *DRM2* genes showed different expression patterns, with one up-regulated, one down-regulated, and one with no significant change. Additionally, the four *PavSUVH5* genes exhibited distinct expression patterns, with one up-regulated, two down-regulated, and one without significant variation. These results suggested that Pd1 dwarfing rootstocks induced changes in the expression of DNA methylation-related genes in “Shuguang2” sweet cherry scions.

### 2.5. Features of the Sweet Cherry Methylome

The methylation status at each cytosine site across the entire genome and the proportion of methylated cytosine sites occurring in each sequence context (CpG, CHH, and CHG (where H represents A, C, or T)) can reflect the characteristics of the genome-wide DNA methylation landscape in a specific species. So, we use whole-genome bisulfite sequencing technology (WGBS) to generate single-base resolution maps of DNA methylation for sweet cherry scions when dwarf plant apical growth was arrested. We collected four groups of samples: shoot tips and leaves from SG-Pd1 (dwarf) (hereafter referred to as Mds_S and Mds_L) and those from SG-WT (vigorous) (hereafter referred to as Mwt_S and Mwt_L), with two biological replicates for each group. We generated up to ~118 million sequencing reads per replicate, yielding 11.83~18.04 Gb of data with Q20 > 94.54% and Q30 > 84.05%. In total, >65.61% of the reads were mapped to the reference genome, and >80.45% of the genome was covered, with an average of >16-fold coverage (Table 1).

As shown in Figure 5A, the sweet cherry genome presented 15.2%~18.6% methylation levels on all sequenced C sites. For different contexts, sweet cherry exhibited the highest methylation levels in the CpG context (59.88~61.55%), the lowest levels in the CHH context (2.99~5.28%), and intermediate levels in the CHG context (28.09~33.78%). Moreover, we analyzed the proportions of mCpG, mCHG, and mCHH at mC sites, which reflected the distribution of mCs in three sequence contexts. As shown in Figure 5B, methylcytosine most frequently appeared at CpG sites (45.34~53.88%), followed by CHG sites (28.95~31.61%) and CHH sites (15.83~21.80%). Compared to SG-WT, the proportion of methylation in symmetric contexts (CpG and CHG) in SG-Pd1 decreased, while the proportion of methylation in the CHH context increased. Comparing methylation levels between shoot tips and leaves, the methylation levels in the three contexts of shoot tips were higher than those of leaves.

We further analyzed the DNA methylation patterns on chromosomes in sweet cherry. From a global perspective of the sweet cherry genome (Figure 5C), we found that the regions with low gene density or high transposable element (TE) content had high CpG and CHG methylation levels. In gene-rich regions, the DNA methylation level in the CpG and CHG contexts was positively correlated with TE density, while the CHH methylation level remained relatively unchanged. In regions with both low gene density and low TE density, DNA methylation levels in the CpG and CHG contexts were high, especially CpG, where methylation reached around 95%, but the CHH methylation level showed a decreasing trend. In addition, the methylation level of each chromosome was evaluated, and we found that chromosome 1 (chr_1) and chromosome 6 (chr_6) had lower CpG and CHG methylation levels and higher CHH methylation levels (Figure 5D). We further analyzed the density distribution of methylated C sites (with methylation level > 0) across different contexts (Figure 5F). CpG and CHG sites generally exhibited high methylation levels, with the majority showing methylation levels ranging from 75% to 100%. In contrast, CHH sites had relatively low methylation levels, with most falling within a range of 10% to 50%.

We further evaluated methylation levels within genes, transposable elements (TEs), and their upstream and downstream 2 kb regions. Each region was divided into 100 bins to calculate the methylation level of each bin. As shown in Figure 5F, for genes, the CpG methylation level was highest and decreased at regions in the vicinity of the transcription start site (TSS) and termination site (TTS). In addition, among CHG and CHH contexts, the methylation level of upstream and downstream regions was higher than that of the body of the gene. RepeatMasker was employed to annotate the TEs in the genome. TEs accounted for 16.86% of the genome, of which 13.39% were retrotransposons. High methylation levels in the CpG and CHG contexts were displayed on TEs, particularly on TE bodies. The methylation level of TE bodies was approximately 91% in the CpG context and about 74% in the CHG context. There was a steady decline on both sides of the TEs, which gradually decreased to around 74% in the CpG context and about 50% in the CHG context. Conversely, the methylation level in the CHH context was much lower on the TEs. In summary, TEs presented higher methylation levels in the CpG and CHG contexts and relatively lower methylation levels in the CHH context.

### 2.6. Dynamic DNA Methylation in Responding to Graft

To investigate the potential effects of dwarfing rootstock on scion methylation, we compared the DNA methylomes of all samples. As shown in Figure 6A, the CHG and CHH methylation levels were increased in Mds_S vs. Mwt_S and Mds_L vs. Mwt_L. Furthermore, we also found that the DNA methylation levels of Mds_S vs. Mwt_S and Mds_L vs. Mwt_L in genes and TEs both had a slight increase, and the trend was consistent in two biological replicates. In addition, among all samples, the DNA methylation levels of Mds_S were highest in the CHG and CHH contexts of genes and in all contexts of TEs (Figure 6B). The methylation levels were used for principal component analysis (PCA), and the results revealed that the same tissue types clustered together in total mC, mCpG, and mCHG sites in PC1. However, for the mCHH sites, Mds_S was distinct from the other three groups, suggesting that CHH methylation significantly changed in shoot tips during grafting (Figure 6C). The above results indicated that Pd1 dwarfing rootstocks led to elevated methylation levels in the scions.

To further investigate the impact of Pd1 dwarfing rootstock on scion DNA methylation changes, differentially methylated regions (DMRs) were identified using a sliding window method with a bin size of 200 bp and a step size of 100 bp. A total of 22,850 DMRs were identified in Mds_S vs. Mwt_S, including 4658 CpG-DMRs, 5049 CHG-DMRs, and 13,143 CHH-DMRs, accounting for 20.38%, 20.10%, and 57.52% of total DMRs, respectively. Meanwhile, 29,516 DMRs were identified in Mds_L vs. Mwt_L, with 11,714 CpG-DMRs, 9182 CHG-DMRs, and 8603 CHH-DMRs, accounting for 39.71%, 31.13%, and 29.16% of total DMRs, respectively. For the CpG-DMRs, the shoot tips and leaves were mostly hypomethylated. For the CHG-DMRs, the shoot tips were mainly hypermethylated, while the leaves were primarily hypomethylated. For the CHH-DMRs, the shoot tips and leaves were mainly hypermethylated (Figure 7A,B).

In total, 16,958 and 9772 hypermethylated DMRs (hyper-DMRs) and 5892 and 19,727 hypomethylated DMRs (hypo-DMRs) in Mds_S vs. Mwt_S and Mds_L vs. Mwt_L were detected, respectively. Our results showed that a larger proportion of DMRs (74.19%) were hypermethylated in Mds_S vs. Mwt_S. In contrast, more DMRs (66.84%) were hypomethylated in Mds_L vs. Mwt_L. To further understand the distribution of DMRs across different sequence contexts, we quantified the number of hyper-DMRs and hypo-DMRs in each sequence context. In Mds_S vs. Mwt_S, CHH hyper-DMRs had the highest number (12,813), accounting for 56.13% of the total DMRs. In Mds_L vs. Mwt_L, CpG/CHG hypo-DMRs (11,219/8497) and CHH hyper-DMRs (8412) were relatively abundant, accounting for 38.01%, 28.11%, and 28.52% of the total DMRs, respectively (Figure 7A,C). Next, we counted the number of DMRs on chromosomes and found that the chromosome with the largest number of DMRs in both the shoot tips and leaves was chromosome 1 (chr1). Furthermore, on chr1, CHH-DMRs were predominant in the shoot tips, whereas CpG-DMRs were dominant in the leaves (Figure 7D). We also analyzed the distribution of DNA methylation changes in DMRs. As shown in Figure 7E, the DNA methylation change levels below the median were relatively concentrated in both shoot tips and leaves, while the change levels above the median exhibited greater variability. The above results suggested that the methylation patterns between shoot tips and leaves in response to grafting were different. Shoot tips from dwarfing plants exhibited a hypermethylated pattern, while leaves exhibited a hypomethylated pattern. The increase in DNA methylation in shoot tips was mostly due to increased CHH methylation.

### 2.7. Differentially Methylated Gene (DMG) Analysis

As shown in Figure 8A, alignment of DMRs with the genome revealed that most DMRs mapped to intergenic and promoter regions. In addition, small amounts of DMRs were found in exon and intron regions. There were 2920 CpG-DMGs, 3455 CHG-DMGs, and 7725 CHH-DMGs in Mds_S vs. Mwt_S, whereas there were 4991 CpG-DMGs, 4328 CHG-DMGs, and 5063 CHH-DMGs in Mds_L vs. Mwt_L. Venn diagram analysis of CpG-DMGs, CHG-DMGs, and CHH-DMGs showed significant overlap across the three clusters, and the proportion of overlapping genes in Mds_S vs. Mwt_S was lower than that in Mds_L vs. Mwt_L, indicating that some DMGs underwent methylation changes in different contexts simultaneously. In addition, there were more DMGs of Mds_L vs. Mwt_L that were simultaneously methylated in two or three sequence contexts. Venn diagram analysis of DMGs in diverse regions demonstrated that most DMGs exhibited methylation alterations only in one region, but a small number of genes underwent methylation alterations simultaneously in multiple regions (Figure 8B). After removing redundancy, a total of 11,197 DMGs were annotated in Mds_S vs. Mwt_S and 10,274 DMGs in Mds_L vs. Mwt_L, of which 5451 DMGs were shared between Mds_S vs. Mwt_S and Mds_L vs. Mwt_L (Figure 8C). To better understand the function of DMGs, GO analysis was used to categorize the unique DMGs to Mds_S vs. Mwt_S, unique DMGs to Mds_L vs. Mwt_L, and common DMGs between Mds_S vs. Mwt_S and Mds_L vs. Mwt_L. As shown in Appendix A, in biological processes, the common DMGs between Mds_S vs. Mwt_S and Mds_L vs. Mwt_L were enriched in meristem maintenance, meristem development, hormone level regulation, hormone metabolic process, phenylpropanoid biosynthesis, gibberellin response, gibberellin biosynthesis, response to wounding, and so on. The DMGs unique to shoot tips were enriched in various processes, such as meristem structural establishment, hormone biosynthetic process, stem system morphogenesis, response to jasmonic acid, meristem initiation, regulation of stem system morphogenesis, auxin response, phenylpropanoid biosynthesis, hormone metabolic process, gibberellin response, gibberellin biosynthesis, and so on. The DMGs unique to leaves were enriched in biological processes like regulation of phosphate metabolic process, hormone transport, meristem development, regulation of hormone levels, and so on. Moreover, a Kyoto Encyclopedia of Genes and Genomes (KEGG) pathway enrichment analysis was performed to further study the biological functions of DMGs. Some of the enriched pathways were the same in Mds_S vs. Mwt_S and Mds_L vs. Mwt_L, notably pathways such as plant hormone signal transduction, the MAPK signaling pathway-plant, phenylpropanoid biosynthesis, and so on (Figure 8D, Appendix A). Further, KEGG pathway enrichment analysis was performed on hyper-DMGs and hypo-DMGs. Hyper-DMGs in Mds_S vs. Mwt_S were significantly enriched in hemiterpene and triterpene biosynthesis, tryptophan metabolism, phenylalanine metabolism, α-linolenic acid metabolism, phenylpropanoid biosynthesis, plant signal transduction, linoleic acid metabolism, and so on. Hypo-DMGs in Mds_S vs. Mwt_S were significantly enriched in plant hormone signal transduction, the MAPK signaling pathway, zeatin biosynthesis, and so on. Hyper-DMGs in Mds_L vs. Mwt_L were enriched in sesquiterpene and triterpenoid biosynthesis, α-linolenic acid metabolism, linoleic acid metabolism, and so on; hypo-DMGs in Mds_L vs. Mwt_L were also enriched in the MAPK signaling pathway-plant, phytohormone signal transduction, phenylalanine metabolism, sesquiterpene and triterpenoid biosynthesis, α-linolenic acid metabolism, linoleic acid metabolism, and so on (Appendix A).

### 2.8. Gene Expression Changes in Dwarfing Scions Caused by Grafting

To obtain accurate transcriptional information of scions grafted onto Pd1 dwarfing rootstocks, transcriptome sequencing was conducted using the same materials as for the WGBS, with three biological replicates. After filtering, approximately 43.44 M and 42.15 M clean reads were obtained from the shoot tips of SG-WT (referred to as WT_S) and SG-Pd1 (referred to as DS_S), respectively. Meanwhile, approximately 43.59 M and 43.70 M clean reads were obtained from the leaves of SG-WT (referred to as WT_L) and SG-Pd1 (referred to as DS_L), respectively (Appendix A). Differentially expressed genes (DEGs) were identified using an error discovery rate (FDR) < 0.05 and a fold change (FC) ≥ 2 as thresholds. As illustrated in Figure 9A, a total of 2198 DEGs were identified in DS_S vs. WT_S, with 1080 up-regulated and 1118 down-regulated. In DS_L vs. WT_L, 1691 DEGs were identified, with 571 up-regulated and 1120 down-regulated. Among these DEGs, the number of DEGs in DS_S vs. WT_S was greater than that in DS_L vs. WT_L. These data indicated that the variety of gene expression in shoot tips was more pronounced than that in leaves in response to graft-mediated dwarfing. As shown in Figure 9B, the Venn diagram showed 1808 unique DEGs in shoot tips, accounting for 82.26%, and 1301 unique DEGs in leaves, accounting for 76.93%. Additionally, 390 DEGs were common to shoot tips and leaves. A volcano plot showing DEGs in DS_S vs. WT_S and DS_L vs. WT_L is shown in Figure 9C.

To explore the functions of DEGs in the shoot tips and leaves, GO and KEGG enrichment analyses were conducted. In biological processes, the DEGs of DS_S vs. WT_S were mainly associated with secondary metabolites, photosynthesis, lignin synthesis, hormones, stress response, plant morphogenesis, and so on. The DEGs of DS_L vs. WT_L were primarily related to lignin synthesis, plant morphogenesis, and so on (Appendix A). KEGG enrichment analysis (Figure 9D, Appendix A) showed that the mainly enriched pathways in DS_S vs. WT_S and DS_L vs. WT_L both included phenylpropanoid biosynthesis, the MAPK signaling pathway-plants, and plant signal transduction pathways, which were also presented in the KEGG analysis of DMGs. In addition, some other KEGG pathways of the DEGs also occurred in those of the DMGs.

### 2.9. Correlation Analysis between DNA Methylation and Gene Expression

We further investigated the relationship between methylation changes and transcriptional alterations in scions under grafting and examined the overlap of DMGs and DEGs. As shown in Figure 10A, most DMGs were not associated with DEGs, and a total of 1020 associated genes were identified in the shoot tips and 828 associated genes in the leaves. In shoot tips, promoter CHH-DMGs (DMGs in the promoter region under the CHH context) with hypermethylation involved the most DEGs, and the numbers of up- and down-regulated DEGs were comparable. In the exon region, most DMGs were upregulated under the CpG context. In addition, intronic CHH-DMGs (DMGs in intronic regions under the CHH context) and intergenic CHH-DMGs (DMGs in intergenic regions under the CHH context) were abundant (Appendix A). In leaves, CpG-DMGs with hypomethylation in the promoter, exon, and intergenic regions were higher than DMGs in other contexts, but in exonic regions the CHH-DMGs were the most abundant. The above results indicate that the genes affected by DNA methylation in scion shoot tips and leaves were quite different (Appendix A). Venn diagram analysis displayed that 130 genes overlapping with DMGs and DEGs were shared between shoot tips and leaves, and 890 unique genes and 697 unique genes were found in shoot tips and leaves, respectively (Figure 10B). Therefore, most of the associated genes between shoot tips and leaves were different, indicating that, in the process of scions responding to dwarfing, the genes affected by methylation were different in the shoot tips and leaves.

Meanwhile, an association analysis between the DNA methylome and the transcriptome was conducted to explore the effect of DNA methylation on gene expression. Genes (tpm > 0) were divided into six groups according to their expression levels, with the 1st quantile (rank1) having the lowest expression level and the sixth quantile (rank6) having the highest expression level. As shown in Figure 10C, in the CpG sequence context, the methylation level of the Rank1 gene group was significantly higher than that of the other five groups in the regions 2 kb upstream and downstream of the gene body. Interestingly, rank1 genes maintained high DNA methylation levels at the TSS and TTS, with noticeable differences in methylation levels before and after these sites. In contrast, the methylation levels of the other five gene groups showed a significant decrease before the TSS and a significant increase after the TSS, peaking in the middle of the gene body and gradually decreasing near the TTS. The decrease near the TTS was less pronounced than that near the TSS, and methylation levels rapidly recovered after the TTS. In the gene body region, the DNA methylation levels of the three low-expression gene groups (rank1~rank3) were lower than those of the three high-expression gene groups (rank4~rank6). This suggests that mCpG in the gene body region enhances gene expression. In the CHG context, the methylation levels of rank1 genes in both the gene body region and the 2 kb upstream and downstream regions were significantly higher than those in the other five groups. In the CHH context, there was a peak in DNA methylation levels before the TSS, which decreased near the TSS. The mCHH level at the region near TSS was significantly positively correlated with gene expression. Next, we determined the differential expression levels of all genes or genes associated with hypomethylated or hypermethylated DMGs. As shown in Figure 10D, the Wilcoxon *p* values of hypomethylated genes compared with all genes were correlated in shoot tips, and those of the other group genes compared with all genes were not correlated. This indicates that many differential transcript abundances were not associated with methylation changes. Together, these data suggest that DNA methylation was at least partially responsible for the transcriptional alterations of these genes.

From the above results of hormone measurements, the dwarfing rootstock affected gibberellin and auxin contents in the scion. Furthermore, the GO and KEGG analyses of DMGs and DEGs indicated that some of the DMGs and DEGs were hormone-related genes. We found some hormone-related genes with significant changes in both methylation and transcription levels, including *SAUR*, *ARF*, *GA2ox*, *ABS1*, and so on (Appendix A). TFs play an important role in regulating gene expression. To characterize the methylation and transcription alterations of TFs, we firstly predicted 2034 TFs from the sweet cherry genome using PlantTFDB. In total, 93 TFs (82 hypermethylated TFs, 33 hypomethylated TFs, and 22 TFs with both hypermethylation and hypomethylation) and 54 TFs (26 hypermethylated TFs, 42 hypomethylated TFs, and 14 TFs with both hypermethylation and hypomethylation) were detected in shoot tips and leaves, respectively (Appendix A), including, notably, *bZIP*, *MYB*, *NAC*, and *AP2*/*ERF* and so on, which are related to plant dwarfing.

## 3. Discussion

Dwarfing rootstocks significantly affect the growth vigor of scions and are widely used in the cultivation of fruit trees, such as cherry, apple, pear, and citrus. In our study, the dwarfing rootstock “Pd1” suppressed the elongation growth of sweet cherry scions. The SAM provides vertical growth for the plant, and the activity of the SAM directly affects the height of the tree [19]. According to the results of the paraffin sectioning analysis, the SAM of the dwarfing plants was smaller and surrounded by lignified scales, and cells in the rib area were arranged tightly, indicating that the cell division ability of the SAM in dwarfing plants was reduced. The evaluation indicators of new branch growth vigor mainly include the length, diameter, and internode length and are the most direct growth indicators for evaluating the degree of tree dwarfing [20,21,22]. In this study, new branches from sweet cherry scions grafted on dwarfing rootstocks displayed shorter lengths, smaller diameters, and shorter internode lengths compared to those grafted on wild-type rootstocks. Our above data indicated that Pd1 dwarfing rootstocks significantly reduced the growth vigor of sweet cherry scions.

WGBS is an especially powerful technology with the ability to determine DNA methylation patterns at single-nucleotide resolution [23]. In animals, methylation occurs only at CpG sites, whereas in plants, all three types of methylation can occur. Previous studies have shown that genome-wide DNA methylation levels in plants vary significantly between species. Niederhuth et al. analyzed 34 angiosperm species and found significant differences in their genome-wide DNA methylation levels [24]. CpG methylation (mCpG) levels varied up to three-fold between different species, ranging from a minimum of approximately 30.5% in *Arabidopsis thaliana* to a high of about 92.5% in *Beta vulgaris*. CHG methylation (mCHG) levels showed great variation among species, with *Eutrema salsugineum* at 9.3% and *B. vulgaris* at 81.2%. CHH methylation (mCHH) levels were generally low and highly variable, ranging from 1.1% in *Vitis vinifera* to 18.8% in *B. vulgaris*. In our study, the levels of mCpG, mCHG, and mCHH in sweet cherry ranged from 59.88% to 61.55%, 28.09% to 33.78%, and 2.99% to 5.28%, respectively. The levels of mCpG, mCHG, and mCHH in sweet cherry are similar to the DNA methylation levels observed in apple, which possesses mCpG (54%), mCHG (38%), and mCHH (8.5%) and also belongs to the Rosaceae family [25]. In this study, we found that CpG methylation levels showed a peak in the gene body and decreased at the regions in the vicinity of the TSS and TTS. Analogous methylation distribution characteristics have been found in other plants like soybean, white birch, and Arabidopsis [26,27,28]. Additionally, in our study, high methylation levels in the CpG and CHG contexts were displayed on TEs, particularly on TE bodies, which was consistent with findings from white birch, poplar, rice, and Arabidopsis [27,28].

An increasing number of studies have shown that grafting can induce DNA methylation changes in scions. Xing et al. used watermelon scions grafted onto gourd rootstocks as materials and found that grafting generally increased the global DNA methylation level of watermelon during the seedling and flowering stages [29]. A past study showed a significant increase in global DNA methylation in cucumber and melon scions grafted onto pumpkin rootstocks [14]. In our study, compared with sweet cherry scions grafted onto wild-type rootstocks (vigorous), the overall genomic methylation level of sweet cherry scions grafted onto Pd1 rootstocks (dwarf) was increased. From the perspective of methylation levels, the methylation level in the CpG context in leaves did not change much, but a large number of CpG-DMRs were identified, indicating that the methylation status in the CpG context in leaves also changed significantly during the process of scion responding to dwarfing rootstocks. In addition, the elevated methylation levels in scions grafted onto Pd1 dwarfing rootstocks were primarily due to an increase in CHH methylation levels. However, the mechanisms by which the rootstock drives methylation changes in the scion remain unclear. Previous studies have shown that mobile small interfering RNAs (siRNAs) between scions and rootstocks can direct DNA methylation [30,31]. Several studies have demonstrated that CHH methylation in plants is a response to stress and plays an important role in responding to environmental stimuli [32,33,34]. Plant grafting can be regarded as a considerable stress [35,36]. Thus, Pd1 dwarfing rootstocks induced elevated methylation levels in sweet cherry scions, probably because grafting acts as a stress to increase the methylation level of CHH, and in addition may be guided by siRNA transported from the rootstock.

In general, DNA methylation inhibits gene expression [37,38]. Nevertheless, a growing number of genome-wide methylome analyses have shown that the relationship between DNA methylation and transcription is more than just a negative correlation. Recently, a study in *Arabidopsis thaliana* clearly indicated that DNA methylation makes a marginal contribution to gene expression [39]. Additionally, Xu et al. found no significant correlation between promoter methylation and gene expression in apples [25]. In particular, in *Populus tomentosa*, highly expressed genes showed higher CpG methylation levels in the gene body [40]. In addition, previous studies in Arabidopsis have indicated that cytosine methylation of the *ROS1* promoter region enhanced the transcription of its downstream genes [41]. In this study, based on the correlation analysis between gene expression and methylation levels, we also found that the high-expression-level genes exhibited higher levels of CpG methylation in the gene body, while the low-expression-level genes displayed high methylation levels at CHG sites in the gene body and 2 kb upstream and downstream. Interestingly, a dramatic increase in CHH methylation around the TSS showed a positive correlation with gene expression. These results suggest that DNA methylation has a complex effect on regulating gene expression in sweet cherry scions grafted onto Pd1 rootstocks (dwarf).

Previous studies have demonstrated that biosynthesis, transport, and signal transduction of plant hormones are regulated by DNA methylation [42,43]. Comparing ovules of high-frequency female-sterile and wild-type rice lines showed that DMGs were mainly involved in plant hormone metabolism [44]. During fruit ripening of *Capsicum annuum*, DNA hypomethylation suppressed the expression of genes related to auxin and gibberellin biosynthesis and cytokinin degradation but evoked the expression of genes related to abscisic acid biosynthesis [45]. Past studies have shown that, in Moso bamboo under abiotic stress, DNA methylation modulated the expression of TF genes and genes related to plant hormone signaling (e.g., *ERF110*, *MYC2*, and *IAA9*) [46]. In our study, DMGs and DEGs were enriched in hormone-related genes. Our data suggest that the alteration of hormone gene expression in sweet cherry scions grafted onto Pd1 rootstocks (dwarf) may be regulated by DNA methylation, thereby affecting hormone synthesis and signal transduction. In addition, methylation variations of TFs affected their expression levels, and dysregulated TFs further altered the transcriptional expression of their downstream target genes [47]. Therefore, the transcription of target genes of TFs may be indirectly dependent on changes in DNA methylation rather than being directly controlled by DNA methylation. For instance, different methylation statuses of the *MYB* promoter resulted in different accumulations of anthocyanins in pear and apple pericarps, leading to varying pericarp colors [48,49]. In this study, we found multiple transcription factors with significant changes in DNA methylation levels, including *NAC*, *MYB*, *WRKY*, *AP2*, *TCP*, and *bHLH*. These TFs are known to be associated with dwarfing, but their functions still need further analysis.

## 4. Materials and Methods

### 4.1. Plant Materials

The scion was dormant buds from the commercial sweet cherry cultivar “Shuguang2 (SG)”. Two types of Tomentosa cherry (*Prunus tomentosa*) rootstock germplasms, wild-type (WT) rootstock (vigorous) and Pd1 rootstock (dwarf), were chosen for grafting. Two different scion/rootstock combinations were used as research objects: “Shuguang2” grafted onto “wild-type rootstock” (SG-WT) and “Shuguang2” grafted onto “Pd1 rootstock” (SG-Pd1). Grafted seedlings were grown in the Wudang area, Guiyang City, Guizhou Province, China (latitude: 26°82′ N and longitude: 106°58′ E).

### 4.2. Morphological Observations of Shoot Tips

The anatomical characteristics of shoot tips were observed using the paraffin slice technique. Shoot tips of sweet cherry scions grafted onto wild-type and dwarfing rootstocks were obtained after 195 days of grafting. Subsequently, the shoot tips were fixed in FAA solution (37% formaldehyde/acetic acid/70% alcohol, 1:1:18, *v*/*v*/*v*), serially dehydrated with different concentrations of ethanol (30–100%), embedded in paraffin, sectioned, and stained with safranin O–fast green. Finally, the stained sections were scanned and analyzed using a Panoramic 250/MIDI scanner with CaseViewer 2.4 software (3DHistech, Budapest, Hungary).

### 4.3. Measurement of Branch Length, Diameter, and Internode Length

After 150 days of grafting, the length and diameter of 15 new branches from sweet cherry scions grafted on wild-type or dwarfing rootstocks were measured randomly, once every 15 days, for a total of five measurements. In addition, we measured the growth vigor of new terminal and new non-terminal branches of four-year-old sweet cherry scions. Preliminary observations found that there was no significant difference in the growth of new branches between the two four-year-old grafted sweet cherries before May. Therefore, in May, we randomly selected and marked 15 new branches with consistent growth vigor from each of the two types of scions on trees at a height of 1.5 m above the ground. Eventually, the length, diameter, and internode length of the marked branches were measured when the leaves fell in November.

### 4.4. Detection of Endogenous GAs and IAA Levels

After 195 days of grafting, shoot tips were collected and used for the detection of endogenous hormone levels. We analyzed gibberellins (GAs) and 3-indoleacetic acid (IAA) levels in shoot tips following the protocol reported by Xin et al. [50]. The phytohormone contents were determined using ultra-performance liquid chromatography coupled with a hybrid triple quadrupole-linear ion trap mass spectrometer (UPLC-QTRAP-MS), which consisted of a Waters UPLC system (Milford, MA, USA) and a Sciex 5500 QTRAP-MS apparatus (Foster City, CA, USA) equipped with an electrospray ionization (ESI) source. A BEH C18 column (1.7 mm, 2.1 mm × 100 mm) was used to separate the phytohormones. The mass spectrum data were processed using Analyst 1.6.2 software (Sciex, Foster City, CA, USA). Finally, the GA and IAA contents were expressed as ng/g fresh weight (Fw).

### 4.5. RT-qPCR

After 195 days of grafting, shoot tip samples were obtained from the sweet cherry scions grafted onto wild-type and dwarfing rootstocks. Subsequently, total RNA was isolated using a plant total RNA extraction kit (Tiangen, Beijing, China) and reverse transcribed to cDNA using a PrimeScript™ RT reagent Kit with gDNA Eraser (TaKaRa, Dalian, China). The qRT-PCR reaction was carried out on a qTOWER3G apparatus (Analytikjena, Jena, Germany) using the PowerUp™ SYBR Green Master Mix (Thermo Fisher, Waltham, MA, USA). Primer sequences are presented in Appendix A. The mRNA expression levels were calculated using delta cycle thresholds (ΔCt), and elongation factor (EF) was used as a reference gene [51].

### 4.6. DNA Extraction and Whole-Genome Bisulfite Sequencing

Referring to the “Standards and Guidelines for Whole Genome Shotgun Bisulfite Sequencing” approved by the National Institutes of Health (NIH) Roadmap Epigenomics Mapping Consortium and the methods reported by Ziller et al., we set up two biological replicates for each group, each replicate with at least 15× coverage [52,53]. After 195 days of grafting, leaves (the third and fourth mature leaves from top to bottom) and shoot tips were obtained from the sweet cherry scions grafted onto wild-type and dwarfing rootstocks. Then, total genome DNA was extracted using a modified CATB (cetyltrimethyl ammonium bromide) method, with two biological replicates for each sample. Subsequently, we measured DNA integrity and concentration using 1% agarose gel electrophoresis and a Nanodrop spectrophotometer (Nanodrop Technologies Inc., Wilmington, NC, USA), respectively. DNA was fragmented using a Bioruptor Pico sonicator (Diagenode, Denville, NJ, USA). Next, dA was added to the 3′ ends, and cytosine-methylated adapters were ligated to the DNA fragments. The ligated DNA fragments were bisulfate-modified using the EZ DNA Methylation-GoldTM kit (Zymo Research Corporation, Irvine, CA, USA). Afterward, we recovered the DNA fragments using the QIAquick Gel Extraction Kit (Qiagen, Valencia, CA, USA) and amplified the DNA fragments using PCR. Finally, qualified libraries were sequenced using BGISEQ-500 with paired-end 100-bp read length.

### 4.7. Whole-Genome Bisulfite Sequencing Data Analysis

Fastp software version 0.20.1 was used to remove adapters, poly-N sequences, and low-quality reads from the raw data to obtain clean data. Then, we used Bismark software (version 0.12.5) with default parameters to compare the clean reads with the reference sweet cherry genome (https://www.rosaceae.org/Analysis/9262820 (accessed on 9 July 2024)) [54]. Differentially methylated regions (DMRs) were identified by the methylKit R package version 1.26.0 using a sliding-window approach with a 200-bp window sliding at 100-bp intervals. DMRs were extracted with a q-value cutoff of 0.01 and a minimum methylation difference of 20% as the threshold. In addition, differentially methylated genes (DMGs) were also characterized.

### 4.8. RNA Extraction and Transcriptome Sequencing

The experimental materials for transcriptome sequencing were the same as those used for whole-genome bisulfite sequencing, and three biological replicates were performed. Total RNA was isolated using a plant total RNA extraction kit (Tiangen, Beijing, China). Afterward, RNA integrity was tested by 1% agarose gel electrophoresis, RNA purity was examined by Nanodrop spectrophotometry, and RNA concentration was determined using a Qubit 2.0 fluorometer (Thermo Fisher Scientific, USA). mRNAs with polyA tails were enriched from total RNA using magnetic beads with oligo-dT tags. Next, the enriched mRNA was fragmented with fragmentation buffer and reverse transcribed into cDNA using N6 random primers. The synthesized cDNA was blunt-ended, phosphorylated at the 5′ end, adenylated at the 3′ end, and ligated to a linker with an overhanging “T”. After PCR amplification, the PCR product was heat-denatured into a single strand, and the single-stranded DNA was circularized with a bridging primer to obtain a single-stranded circular DNA library. Subsequently, initial quantification was performed using a Qubit 2.0 fluorometer, the library was diluted to 1.5 ng/μL, and an Agilent 2100 instrument was used to detect the size of the inserted fragment. Finally, the constructed library was sequenced using the BGISEQ-500 platform (Huada Gene Technology, Shenzhen, China) (150 bp paired-end reads).

### 4.9. RNA-Seq Data Analysis

Clean data were acquired by removing adapters, poly-N sequences, and low-quality reads from the raw data using SOAPnuke software version 1.5.6. Then, clean reads were aligned to the reference sweet cherry genome (https://www.rosaceae.org/Analysis/9262820 (accessed on 9 July 2024)) [54] using Hisat2 software version 2.2.1 with default parameters. Differentially expressed genes (DEGs) were identified by the DESeq R package version 1.40.2. A false discovery rate (FDR) of less than 0.05 and an absolute fold change (FC) value of ≥2 were selected as limits for determining whether differential gene expression was significant.

### 4.10. Principal Component Analysis and Enrichment Analysis

Principal component analysis (PCA) was performed using the PCAtools package (version 3.6.1). In addition, GO (Gene Ontology) and KEGG (Kyoto Encyclopedia of Genes and Genomes) enrichment analyses were conducted using the R package ClusterProfiler version 4.8.3.

## 5. Conclusions

Pd1 rootstock reduced the cell division ability of the SAM and decreased the levels of GAs and IAA in sweet cherry scions. Methylome analysis showed that the sweet cherry genome presented 15.2~18.6%, 59.88~61.55%, 28.09~33.78%, and 2.99~5.28% methylation levels at all sequenced C sites and CG, CHG, and CHH contexts, respectively. Shoot tips from dwarfing plants exhibited a hypermethylated pattern, while leaves exhibited a hypomethylated pattern. The DNA methylation increase in shoot tips was mostly due to increased CHH methylation. GO and KEGG analyses showed that DMGs and DEGs were enriched in hormone-related GO terms and KEGG pathways. Global methylation and transcription analysis revealed that mCpG in the gene body region enhanced gene expression, and mCHH at the region near TSS was positively correlated with gene expression. Many DEGs were not associated with the methylation changes. In addition, some hormone-related genes and TFs had significant changes in methylation and transcription levels, including *SAUR*, *ARF*, *GA2ox*, *ABS1*, *bZIP*, *MYB*, *NAC*, and so on, which are known to be involved in plant dwarfing. These findings can lay a good basis for further epigenetic studies on sweet cherry dwarfing and provide valuable new insight into understanding rootstock–scion interactions.

## Figures and Tables

**Figure 1 ijms-25-11100-f001:**
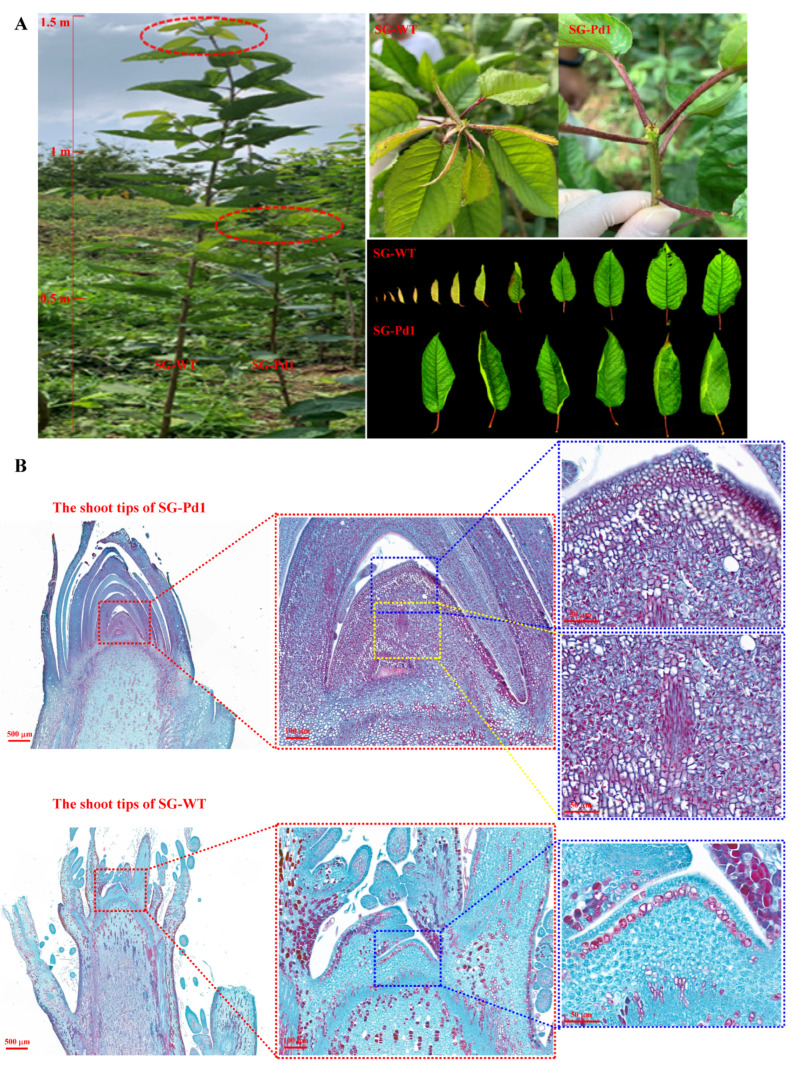
Growth and anatomical observation of shoot tips. (**A**) Grafting plants at the growth stagnation stage (195 days after grafting). Red circles mark shoot tips. (**B**) Anatomical observation of shoot tips. SG-Pd1: “Shuguang2” sweet cherry grafted onto Pd1 rootstock (dwarf); SG-WT: “Shuguang2” sweet cherry grafted onto wild-type rootstock (vigorous).

**Figure 2 ijms-25-11100-f002:**
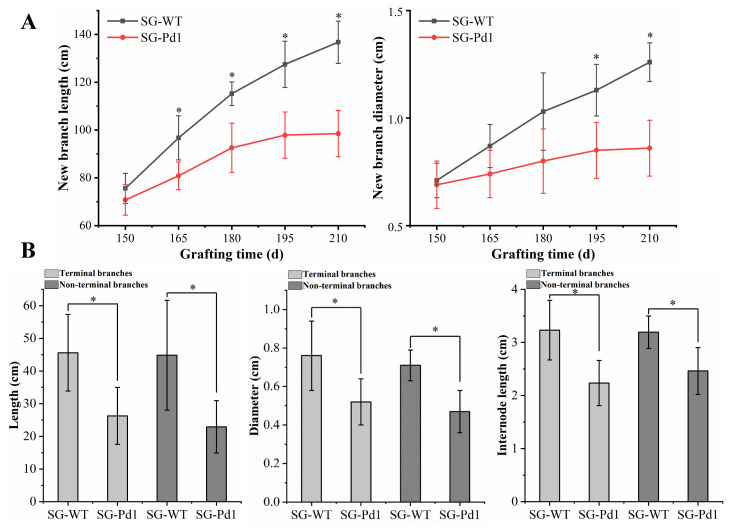
The effect of rootstock on scion growth. (**A**) The growth of “Shuguang2” grafted buds. (**B**) New branch growth of 4-year-old grafted plants. Significant differences from the fifteen biological replicates were calculated using *t*-test and indicated by * (*p* value < 0.05). SG-Pd1: “Shuguang2” sweet cherry grafted onto Pd1 rootstock (dwarf); SG-WT: “Shuguang2” sweet cherry grafted onto wild-type rootstock (vigorous).

**Figure 3 ijms-25-11100-f003:**
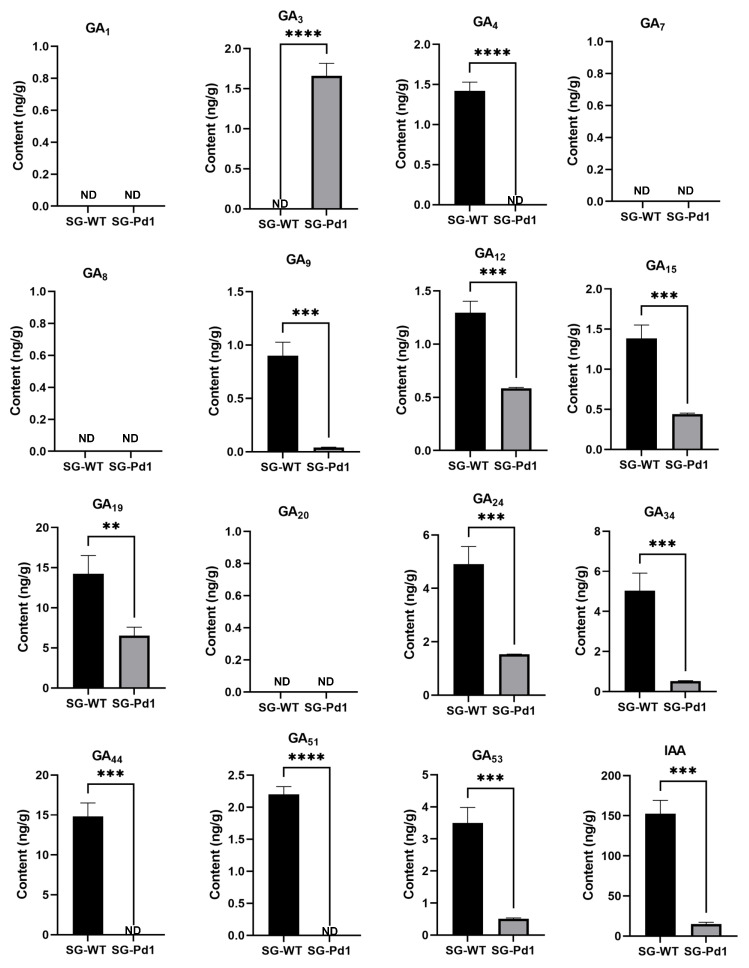
The content of GAs and IAA in shoot tips. ND indicates “not detected”. Significant differences were evaluated using *t*-test and indicated by ** (*p* value < 0.01), *** (*p* value < 0.001), and **** (*p* value < 0.0001). SG-Pd1: “Shuguang2” grafted onto Pd1 rootstock (dwarf); SG-WT: “Shuguang2” grafted onto wild-type rootstock (vigorous).

**Figure 4 ijms-25-11100-f004:**
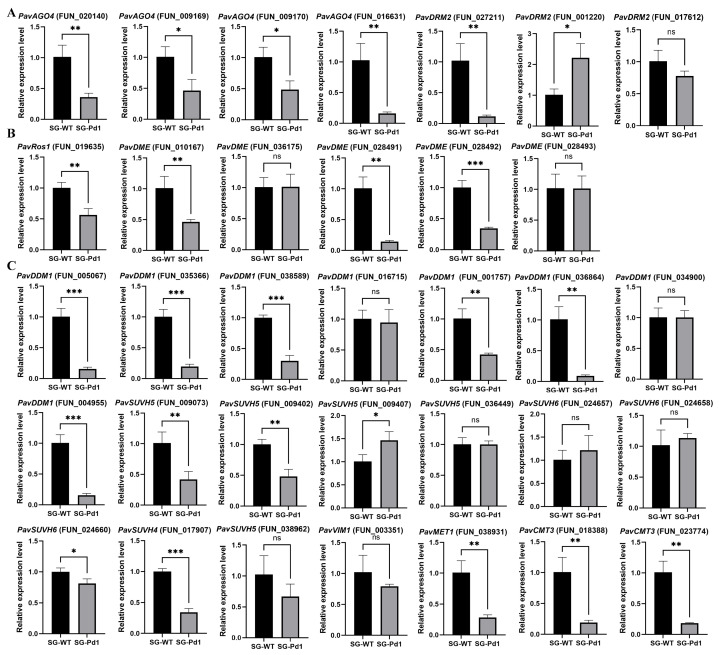
Relative expression analysis of genes encoding DNA methylation-related enzymes using RT-qPCR. (**A**) Genes encoding de novo methylation-related enzymes. (**B**) Genes encoding demethylation-related enzymes. (**C**) Genes encoding methylation maintenance-related enzymes. Significant differences were assessed by *t*-test and are indicated by * (*p* value < 0.05), ** (*p* value < 0.01), and *** (*p* value < 0.001), with ns indicating “not significant”.

**Figure 5 ijms-25-11100-f005:**
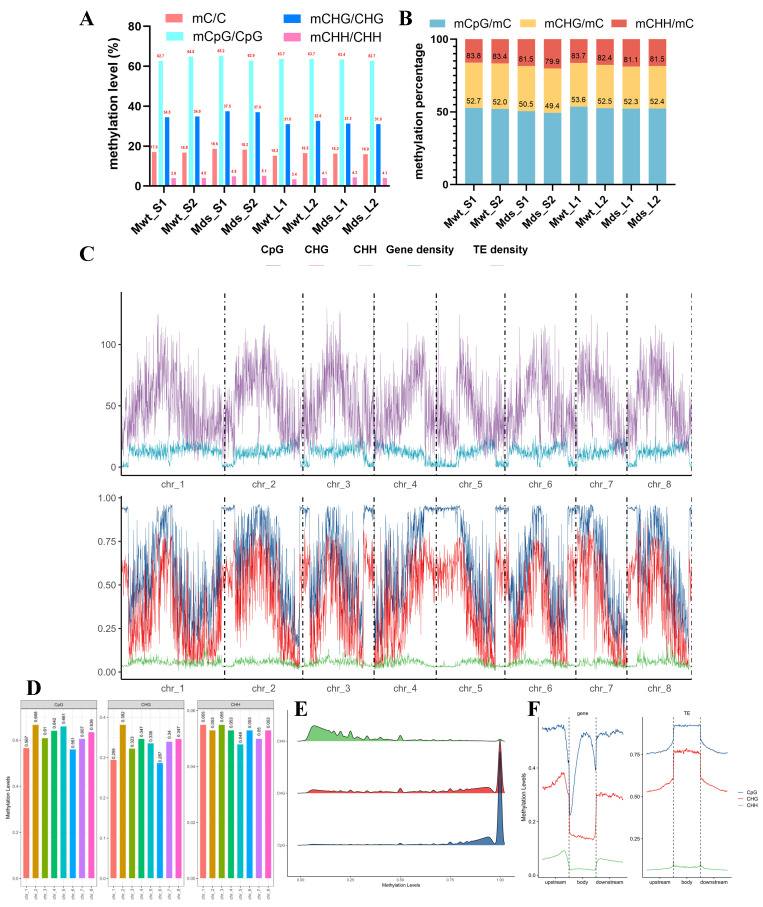
Features of the sweet cherry scion methylome. (**A**) The methylation level of sweet cherry scions. (**B**) Relative proportions of mCpG, mCHG, and mCHH in total mC. (**C**) DNA methylation of chromosomes in the sweet cherry genome. (**D**) The methylation level of each chromosome in sweet cherry. (**E**) Methylation density. (**F**) The DNA methylation features of genes and TEs. Mds_S and Mds_L indicated shoot tips and leaves from SG-Pd1, respectively; Mwt_S and Mwt_L indicated shoot tips and leaves from SG-WT, respectively.

**Figure 6 ijms-25-11100-f006:**
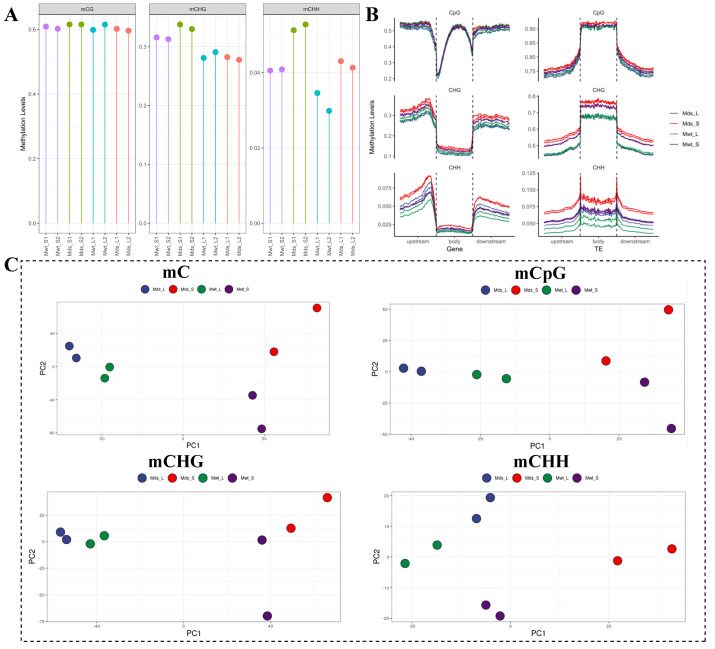
Dynamic changes in DNA methylation after grafting. (**A**) The DNA methylation level of each sample. (**B**) DNA methylation characteristics of genes and TEs. (**C**) Principal component analysis (PCA) of total mC, mCpG, mCHG, and mCHH. Mds_S and Mds_L indicate shoot tips and leaves from SG-Pd1, respectively; Mwt_S and Mwt_L indicate shoot tips and leaves from SG-WT, respectively.

**Figure 7 ijms-25-11100-f007:**
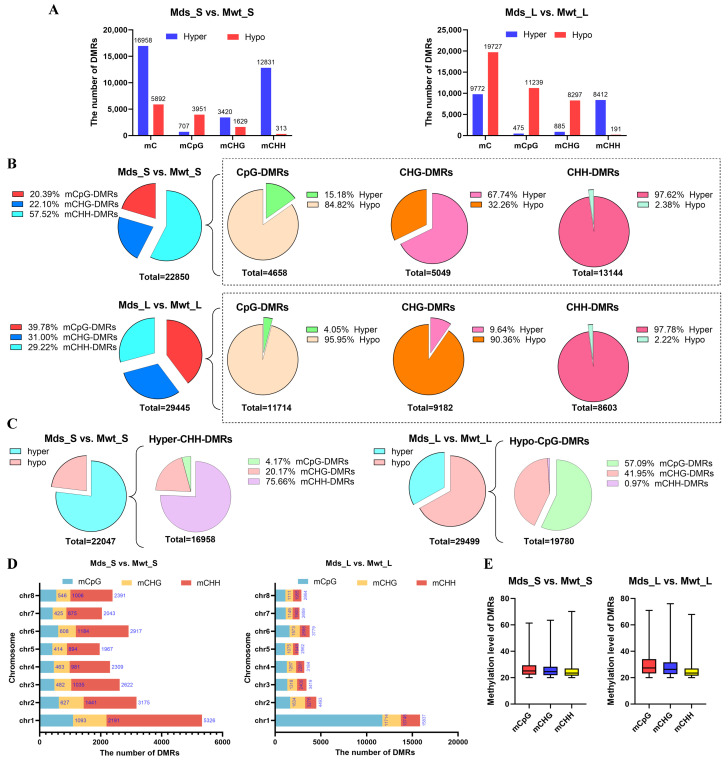
Identification of DMRs. (**A**) The number of DMRs identified in Mds_S vs. Mwt_S (shoot tips from SG-Pd1 vs. shoot tips from SG-WT) and Mds_L vs. Mwt_L (leaves from SG-Pd1 vs. leaves from SG-WT). (**B**) The proportion of CpG-DMRs, CHG-DMRs, and CHH-DMRs. (**C**) The proportion of hyper-DMRs and hypo-DMRs. (**D**) The number of DMRs on chromosomes (chr1–chr8). (**E**) Boxplot of DNA methylation changes in DMRs.

**Figure 8 ijms-25-11100-f008:**
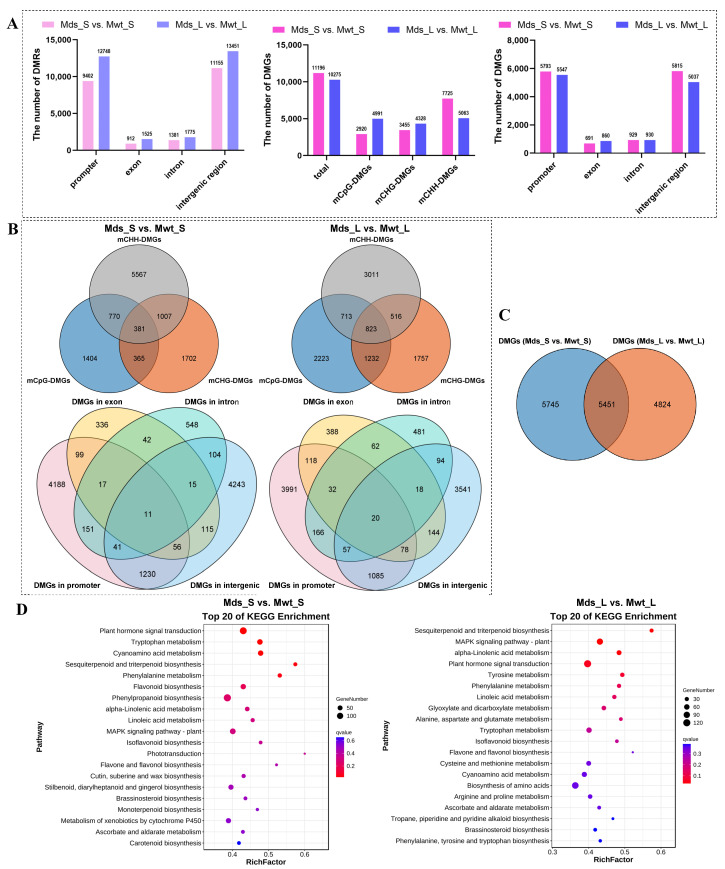
Differentially methylated gene (DMG) analysis. (**A**) The number of DMGs identified in Mds_S vs. Mwt_S (shoot tips from SG-Pd1 vs. shoot tips from SG-WT) and Mds_L vs. Mwt_L (leaves from SG-Pd1 vs. leaves from SG-WT). (**B**) Venn diagram analysis of DMGs in different sequence contexts (mCpG-DMGs, mCHG-DMGs, and mCHH-DMGs) and regions (promoter, exon, and intronic and intergenic regions). (**C**) Venn diagram analysis of DMGs. (**D**) KEGG pathway enrichment analysis of DMGs.

**Figure 9 ijms-25-11100-f009:**
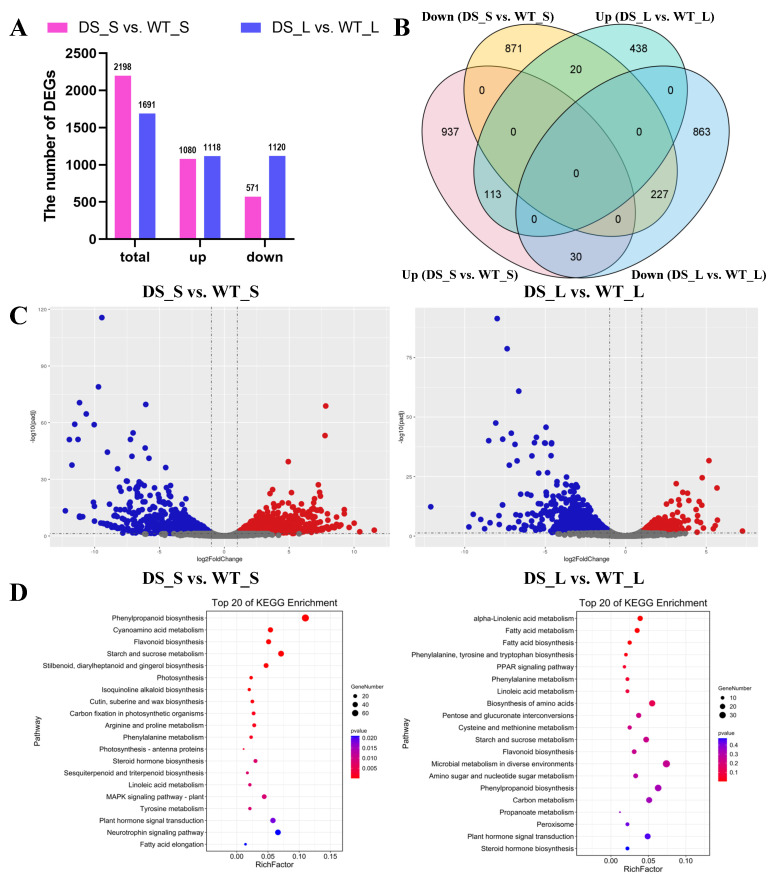
Differentially expressed gene (DEGs) analysis. (**A**) The number of DEGs. (**B**) Venn diagram of DEGs in shoot tips and leaves. (**C**) Volcano plot showing DEGs between shoot tips and leaves. (**D**) KEGG enrichment analysis was separately conducted on the DEGs in the shoot tips and leaves. WT_S and DS_S indicate shoot tips of SG-WT and SG-Pd1, respectively; WT_L and DS_L indicate leaves of SG-WT and SG-Pd1, respectively.

**Figure 10 ijms-25-11100-f010:**
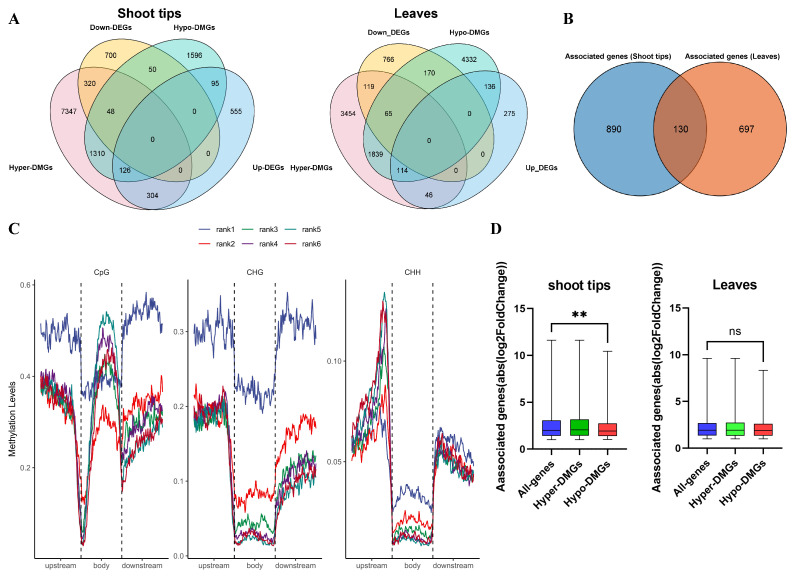
Correlation analysis between DNA methylation and gene expression. (**A**) Venn diagram analysis of DEGs and DMGs in shoot tips and leaves. (**B**) Venn diagram analysis of overlapping genes in shoot tips and leaves. (**C**) Distributions of methylation levels within genes partitioned by different expression levels: rank1 is the lowest, and rank6 is the highest. (**D**) Differential expression levels of all genes, hypermethylated genes, and hypomethylated genes, displayed as boxplots. Wilcoxon *p* values indicated by ** (*p* value < 0.005). In addition, ns indicates “not significant”.

**Table 1 ijms-25-11100-t001:** Data description of WGBS reads for the four group samples with two replicates.

Samples	Clean Reads/Data Production (Gb)	Q20	Q30	Mapping Reads	Mapping Rate	Average Coverage Rate	Bisulfite Conversion Rate (%)	Average Depth (X)
Mwt_S1	118,321,284/11.82	94.54%	84.05%	59,155,804	69.76%	82.21%	99.36	16.8
Mwt_S2	118,809,618/11.88	96.79%	86.37%	81,568,688	77.51%	83.72%	99.44	25.6
Mds_S1	136,310,954/13.63	95.09%	86.54%	68,150,418	68.71%	80.45%	99.36	18.1
Mds_S2	152,574,296/15.26	95.67%	86.84%	76,283,514	72.92%	83.94%	99.42	22.6
Mwt_L1	156,000,486/15.60	96.68%	89.25%	77,997,226	74.40%	83.11%	99.44	23.1
Mwt_L2	180,352,376/18.04	96.27%	88.18%	76,234,395	74.4%	84.46%	99.44	24.6
Mds_L1	132,832,352/13.28	96.33%	88.44%	66,413,378	65.61%	81.72%	99.39	17.5
Mds_L2	152,475,654/15.25	96.02%	87.63%	72,554,094	71.58%	83.42%	99.43	21.3

## Data Availability

The data presented in this study are available on request from the corresponding author.

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
