# Peer review of "Single-Base Methylome Analysis of Sweet Cherry (*Prunus avium* L.) on Dwarfing Rootstocks Reveals Epigenomic Differences Associated with Scion Dwarfing Conferred by Grafting"

_ijms, 2024, doi:10.3390/ijms252011100_

Round 1
Reviewer 1 Report
Comments and Suggestions for Authors
In the manuscript that was submitted by Xiaopeng Wen and colleagues, entitled ‘Single-base Methylome Analysis of Sweet Cherry (Prunus avium. L) on Dwarfing Rootstocks Reveals Epigenomic Differences Associated with Scions Dwarfing Conferred by Grafting’, the authors study the dwarfing on sweet cherry scion using quantification of plant growth regulators and integration analysis of epigenome and transcriptome.
Among the positive aspects of the manuscript are: (a) the writing is fine, (b) the figures have a good presentation, (c) the purpose is clear. The negatives of the manuscript include; (a) additional samples should be run for WGBS analysis, (b) reanalysis of WGBS data should be done. Overall, the manuscript cannot be accepted for publication with only 2 biological replicates for the estimation of methylated cytosines per tissue / rootstock in IJMS.
Major and minor
Lines 126 and 128; the expression ‘downregulated’ is it for gene expression, in metabolites, it is not accurate; you should use ‘decreased their abundance’.
Line 132, Figure Legend: Provide the test (t-test or other) you use to perform statistical analysis. (similar line 154, figure legend)
Lines 145 and 146; You should also comment PavSUVH5 gene, where 1 up, 2 down and 1 ns (Figure 4).
Major concern. Why do you use 2 biological replications instead of 3 biological replications to do WGBS analysis? My main concern is about the number of replications and how, after that find DMRs significant. In my opinion, the WGBS should be done with 3 biological replications (at least) in each tissue or rootstock.
In these kinds of experiments, they usually use grafting a scion into oneself as a control. Why did you choose a wild type? Please explain.
Author Response
Single-base Methylome Analysis of Sweet Cherry (Prunus avium L.) on Dwarfing Rootstocks Reveals Epigenomic Differences Associated with Scions Dwarfing Conferred by Grafting
(Manuscript ID: ijms-3175946)
Dear Reviewer:
Thanks for your comments concerning our manuscript entitled “Single-base Methylome Analysis of Sweet Cherry (Prunus avium L.) on Dwarfing Rootstocks Reveals Epigenomic Differences Associated with Scions Dwarfing Conferred by Grafting” (Manuscript ID: ijms-3175946). Those comments are all valuable and very helpful for revising and improving our paper, as well as the important guiding significance to our researches. We have studied comments carefully and have made correction which we hope meet with approval. Revised portion are marked in red in the paper. The main corrections in the paper and the responds to the reviewer’s comments are as flowing:
Responds to the reviewer’s comments:
For the Reviewer 1:
Comments 1: [In the manuscript that was submitted by Xiaopeng Wen and colleagues, entitled ‘Single-base Methylome Analysis of Sweet Cherry (Prunus avium. L) on Dwarfing Rootstocks Reveals Epigenomic Differences Associated with Scions Dwarfing Conferred by Grafting’, the authors study the dwarfing on sweet cherry scion using quantification of plant growth regulators and integration analysis of epigenome and transcriptome.]
Response 1: Thank you very much for the positive comments on our work.
Comments 2: [Among the positive aspects of the manuscript are: (a) the writing is fine, (b) the figures have a good presentation, (c) the purpose is clear. The negatives of the manuscript include; (a) additional samples should be run for WGBS analysis, (b) reanalysis of WGBS data should be done. Overall, the manuscript cannot be accepted for publication with only 2 biological replicates for the estimation of methylated cytosines per tissue / rootstock in IJMS.]
Suggestion 2.1: [Among the positive aspects of the manuscript are: (a) the writing is fine, (b) the figures have a good presentation, (c) the purpose is clear.]
Response 2.1: Thank you very much for the positive comments on our work.
Suggestion 2.2: [The negatives of the manuscript include; (a) additional samples should be run for WGBS analysis, (b) reanalysis of WGBS data should be done. Overall, the manuscript cannot be accepted for publication with only 2 biological replicates for the estimation of methylated cytosines per tissue / rootstock in IJMS.]
Response 2.2: During the DNA extraction process, we performed three biological replicates. However, the amount of DNA in some samples from both shoot tip and leaf tissues did not meet the requirements for library construction. In addition, we reviewed relevant literature, such as Cheng et al., 2018; Huang et al., 2019; Liu et al., 2023a; Liu et al., 2023b; Rawoof et al., 2020; Zhang et al., 2020. Based on the methods in these references, two biological replicates or no biological replicates were used in WGBS analysis. Therefore, we used two biological replicates in the WGBS analysis. Thank you very much for the positive comments on our work.
References
Cheng. J.; Niu, Q.; Zhang, B.; Chen, K.; Yang, R.; Zhu, J.K.; Zhang, Y.; Lang, Z. Downregulation of RdDM during strawberry fruit ripening. Genome Biol. 2018, 19, 212.
Huang, H.; Liu, R.; Niu, Q.; Tang, K.; Zhang, B.; Zhang, H.; Chen, K.; Zhu, J.K.; Lang, Z. Global increase in DNA methylation during orange fruit development and ripening. PNAS 2019, 22, 116, 1430-1436.
Liu, H.N.; Shu, Q.; Lin-Wang, K.; Espley, R.V.; Allan, A.C.; Pei, M.S.; Li, X.L.; Su, J.; Wu, J. DNA methylation reprogramming provides insights into light-induced anthocyanin biosynthesis in red pear. Plant Sci. 2023a, 326, 111499.
Liu, Y.; Priyadarshani, S.V.G.N.; Chi, M.; Yan, M.; Mohammadi, M.A., Zhang, M.; Zhou, Q.; Wang, L.; Luo, T.; Wai, M.H.; Wang, X.; Cai, H.; Wang, H.; Qin, Y. Epigenetic modification mechanisms of chloroplasts mutants in pineapple leaves during somatic regeneration. Hortic. Plant J. 2023b, 9, 509-522.
Rawoof, A.; Chhapekar, S.S.; Jaiswal, V.; Brahma, V.; Kumar, N.; Ramchiary, N. Single-base cytosine methylation analysis in fruits of three Capsicum species. Genomics 2020, 112, 3342-3353.
Zhang, M.; Zhang, X.; Guo, L.; Qi, T.; Liu, G.; Feng, J.; Shahzad, K.; Zhang, B.; Li, X.; Wang, H.; Tang, H.; Qiao, X.; Wu, J.; Xing, C. Single-base resolution methylome of cotton cytoplasmic male sterility system reveals epigenomic changes in response to high-temperature stress during anther development. J. Exp. Bot. 2020, 23, 71, 951-969.
Comments 3: [Lines 126 and 128; the expression ‘downregulated’ is it for gene expression, in metabolites, it is not accurate; you should use ‘decreased their abundance’.]
Response 3: In line 18, we have replaced “downregulated” with “decreased”. In line 128, “downregulated” has been replaced by “decreased their abundance”. In line 130, we have replaced “downregulated” with “reduced”. In line 648, we have replaced “downregulated” with “decreased”. Revised portions are marked in red in the paper. Thank you for these positive and constructive comments and suggestions.
Comments 4: [Line 132, Figure Legend: Provide the test (t-test or other) you use to perform statistical analysis. (similar line 154, figure legend)]
Response 4: In lines 119-120 (Figure 2 Legend), “Significant differences calculated from the fifteen biological replicates were indicated by * (P value < 0.05).” has been replaced by “Significant differences from the fifteen biological replicates were calculated using t-test and indicated by * (P value < 0.05).”. In lines 133-135 (Figure 3 Legend), “Significant differences were indicated by *** (P value < 0.001) and **** (P value < 0.0001).” has been replaced by “Significant differences were evaluated using t-test and indicated by *** (P value < 0.001) and **** (P value < 0.0001).”. In lines 157-159 (Figure 4 Legend), “Significant differences were indicated by * (P value < 0.05), ** (P value < 0.01), and *** (P value < 0.001)” has been replaced by “Significant differences were performed using t-test and indicated by * (P value < 0.05), ** (P value < 0.01), and *** (P value < 0.001).”. Revised portions are marked in red in the paper. Thank you for these positive and constructive comments and suggestions.
Comments 5: [Lines 145 and 146; You should also comment PavSUVH5 gene, where 1 up, 2 down and 1 ns (Figure 4).]
Response 5: In lines 148-151, “Additionally, the four PavSUVH5 genes exhibited distinct expression patterns, with one up-regulated, two down-regulated, and one without significant variation.” has been added. Revised portions are marked in red in the paper. Thank you for these positive and constructive comments and suggestions.
Comments 6: [Why do you use 2 biological replications instead of 3 biological replications to do WGBS analysis? My main concern is about the number of replications and how, after that find DMRs significant. In my opinion, the WGBS should be done with 3 biological replications (at least) in each tissue or rootstock.]
Response 6: During the DNA extraction process, we performed three biological replicates. However, the amount of DNA in some samples from both shoot tip and leaf tissues did not meet the requirements for library construction. In addition, we reviewed relevant literature, such as Cheng et al., 2018; Huang et al., 2019; Liu et al., 2023a; Liu et al., 2023b; Rawoof et al., 2020; Zhang et al., 2020. Based on the methods in these references, two biological replicates or no biological replicates were used in WGBS analysis. Therefore, we used two biological replicates in the WGBS analysis. Thank you very much for the positive comments on our work.
References
Cheng. J.; Niu, Q.; Zhang, B.; Chen, K.; Yang, R.; Zhu, J.K.; Zhang, Y.; Lang, Z. Downregulation of RdDM during strawberry fruit ripening. Genome Biol. 2018, 19, 212.
Huang, H.; Liu, R.; Niu, Q.; Tang, K.; Zhang, B.; Zhang, H.; Chen, K.; Zhu, J.K.; Lang, Z. Global increase in DNA methylation during orange fruit development and ripening. PNAS 2019, 22, 116, 1430-1436.
Liu, H.N.; Shu, Q.; Lin-Wang, K.; Espley, R.V.; Allan, A.C.; Pei, M.S.; Li, X.L.; Su, J.; Wu, J. DNA methylation reprogramming provides insights into light-induced anthocyanin biosynthesis in red pear. Plant Sci. 2023a, 326, 111499.
Liu, Y.; Priyadarshani, S.V.G.N.; Chi, M.; Yan, M.; Mohammadi, M.A., Zhang, M.; Zhou, Q.; Wang, L.; Luo, T.; Wai, M.H.; Wang, X.; Cai, H.; Wang, H.; Qin, Y. Epigenetic modification mechanisms of chloroplasts mutants in pineapple leaves during somatic regeneration. Hortic. Plant J. 2023b, 9, 509-522.
Rawoof, A.; Chhapekar, S.S.; Jaiswal, V.; Brahma, V.; Kumar, N.; Ramchiary, N. Single-base cytosine methylation analysis in fruits of three Capsicum species. Genomics 2020, 112, 3342-3353.
Zhang, M.; Zhang, X.; Guo, L.; Qi, T.; Liu, G.; Feng, J.; Shahzad, K.; Zhang, B.; Li, X.; Wang, H.; Tang, H.; Qiao, X.; Wu, J.; Xing, C. Single-base resolution methylome of cotton cytoplasmic male sterility system reveals epigenomic changes in response to high-temperature stress during anther development. J. Exp. Bot. 2020, 23, 71, 951-969.
Comments 7: [In these kinds of experiments, they usually use grafting a scion into oneself as a control. Why did you choose a wild type? Please explain.]
Response 7: In our previous breeding process, we attempted to cultivate seedlings using seeds from the sweet cherry cultivar ‘Shuguang2 (SG)’, but the seeds failed to germinate. Hence, we were unable to obtain seedlings for self-grafting. In this work, we aim to explore the effects of dwarfing rootstock 'Pd1' on sweet cherry ‘Shuguang2’ scion using quantification of plant growth regulators and integration analysis of epigenome and transcriptome. Thus, two types of Tomentosa cherry (Prunus tomentosa) rootstock germplasms, wild-type (WT) rootstock (vigorous) and Pd1 rootstock (dwarf), were chosen for grafting. Thank you very much for the positive comments on our work.
We would like to express our great appreciation to you for comments on our paper. Looking forward to hearing from you.
Thank you and best regards.
Yours sincerely,
Corresponding author: Xiaopeng Wen
wenxp@gzu.edu.cn (X.W.)

Reviewer 2 Report
Comments and Suggestions for Authors
Dear Authors, below are my comments:
Abstract: It is appropriate to maximize this section to the 200 words defined by the MDPI. It is advisable not to cite other, previous work. More detail should be provided on the purpose of the work, including the material and method used - this is not currently available. It is also recommended that the conclusion be slightly highlighted.
Key words: understated and small in number. It is worth choosing different keywords than those in the title and specialising this a bit.
Introduction: I suggest that the purpose of the work be highlighted more.
Discussion: I miss the deep comparison of the literature reviewed and my own results, I suggest rewriting this chapter.
Conclusions: it would be useful to include the more distant aims of the work and its impact on horticulture and agriculture, I suggest rewriting this section.
- The main question addressed by the research about Prunus avium inoculation due to DNA methylation and gene expression is presented in the manuscript as a possible correlation.
- The topic original or relevant to the field with increasing climate change, we need to pay a lot of attention to the production of food crops and to present other new methods - and this manuscript is a good example.
- It works on a classic topic in the context of genetic modification and DNA methylation. - In the review I wrote: keywords, abstract, discussion, conclusion chapters should be modified. - A deeper discussion is needed between own literature and previous publications.Author Response
Single-base Methylome Analysis of Sweet Cherry (Prunus avium L.) on Dwarfing Rootstocks Reveals Epigenomic Differences Associated with Scions Dwarfing Conferred by Grafting
(Manuscript ID: ijms-3175946)
Dear Reviewer:
Thanks for your comments concerning our manuscript entitled “Single-base Methylome Analysis of Sweet Cherry (Prunus avium L.) on Dwarfing Rootstocks Reveals Epigenomic Differences Associated with Scions Dwarfing Conferred by Grafting” (Manuscript ID: ijms-3175946). Those comments are all valuable and very helpful for revising and improving our paper, as well as the important guiding significance to our researches. We have studied comments carefully and have made correction which we hope meet with approval. Revised portion are marked in red in the paper. The main corrections in the paper and the responds to the reviewer’s comments are as flowing:
Responds to the reviewer’s comments:
For the Reviewer 2:
Comments 1: [Abstract: It is appropriate to maximize this section to the 200 words defined by the MDPI. It is advisable not to cite other, previous work. More detail should be provided on the purpose of the work, including the material and method used - this is not currently available. It is also recommended that the conclusion be slightly highlighted.]
Response 1: In abstract, relevant descriptions of our previous work “In our previous work, a dwarf rootstock germplasm 'Pd1' was obtained from the diverse germplasm of Tomentosa cherry (Prunus tomentosa) in Guizhou Province, China. Compared with the wild type, Pd1 exhibited an excellent dwarfing effect on sweet cherry scions.” have been deleted. In addition, we have added more detail on the purpose of the work, including the material and method used. We have also highlighted the conclusion. Revised portions are marked in red in the paper. Thank you very much for the positive comments on our work.
Currently revised abstract:
Plant grafting using dwarfing rootstocks is one of the important cultivation measures in the sweet cherry (Prunus avium) industry. In this work, we aimed to explore the effects of dwarfing rootstock 'Pd1' (Prunus tomentosa) on sweet cherry ‘Shuguang2’ scion by performing morphological observations using paraffin slice techniques, detecting GA and IAA contents using UPLC-QTRAP-MS, and implementing integration analyses of epigenome and transcriptome using whole-genome bisulfite sequencing and transcriptome sequencing. Anatomical analysis indicated that the cell division ability of the SAM in dwarfing plants was reduced. Pd1 rootstock significantly decreased the levels of GAs and IAA in sweet cherry scions. Methylome showed the sweet cherry genome presented 15.2%~18.6%, 59.88%~61.55%, 28.09%~33.78%, and 2.99%~5.28% methylation at the total C, CG, CHG, and CHH sites, respectively. Shoot tips from dwarfing plants exhibited a hypermethylated pattern mostly due to increased CHH methylation, while leaves exhibited a hypomethylated pattern. According to the GO and KEGG analysis, DMGs and DEGs were enriched in hormone-related GO terms and KEGG pathways. Global correlation analysis between methylation and transcription revealed that the mCpG in the gene body region enhanced gene expression, and the mCHH at the region near TSS was positively correlated with gene expression. Next, we found some hormone-related genes and TFs with significant changes in methylation and transcription, including SAURs, ARF, GA2ox, ABS1, bZIP, MYB, and NAC. This study presented a methylome map of the sweet cherry genome, revealed scion widespread DNA methylation alterations caused by dwarfing rootstock, and obtained abundant genes with methylation and transcription alterations that were potentially involved in rootstock-induced growth changes in the sweet cherry scion. Our findings can lay a good basis for further epigenetic studies on sweet cherry dwarfing and provide valuable new insight into understanding rootstock-scion interactions.
Comments 2: [Key words: understated and small in number. It is worth choosing different keywords than those in the title and specialising this a bit.]
Response 2: The key words “Sweet cherry; graft; dwarf; DNA methylation” has been replaced by “Prunus tomentosa; DNA methylation; whole-genome bisulfite sequencing; transcriptome; rootstock-scion interactions”. Revised portions are marked in red in the paper. Thank you for these positive and constructive comments and suggestions.
Comments 3: [Introduction: I suggest that the purpose of the work be highlighted more.]
Response 3: In the introduction (lines 79-80), we have highlighted the purpose of our work. “In the present study, we aimed to explore the potential epigenetic regulatory of Pd1 dwarfing rootstock's influence on sweet cherry scion vigor.” has been added. Revised portions are marked in red in the paper. Thank you for these positive and constructive comments and suggestions.
Comments 4: [Discussion: I miss the deep comparison of the literature reviewed and my own results, I suggest rewriting this chapter.]
Response 4: In the discussion, we have provided a more in-depth discussion based on our own research and previous publications. Revised portions are marked in red in the paper. Thank you for these positive and constructive comments and suggestions.
Currently added references:
- An, Y.C.; Goettel, W.; Han, Q.; Bartels, A.; Liu, Z.; Xiao, W. Dynamic changes of genome-wide DNA methylation during soybean seed development. Rep. 2017, 7, 12263.
- Su, C.; Wang, C.; He, L.; Yang, C.; Wang, Y. Shotgun bisulfite sequencing of the Betula platyphylla genome reveals the tree’s DNA methylation patterning. J. Mol. Sci. 2014, 15, 22874–22886.
- Feng, S.; Cokus, S.J.; Zhang, X.; Chen, P.; Bostick, M.; Goll, M.G.; Hetzel, J.; Jain, J.; Strauss, S.H.; Halpern, M.E.; et al. Conservation and divergence of methylation patterning in plants and animals. Natl. Acad. Sci. USA 2010, 107, 8689–8694.
- Molnar, A.; Melnyk, C.W.; Bassett, A.; Hardcastle, T.J.; Dunn, R.; Baulcombe, D.C. Small silencing RNAs in plants are mobile and direct epigenetic modification in recipient cells. Science 2010, 328, 872–875.
- Bai, S.; Kasai, A.; Yamada, K.; Li, T.; Harada, T. A mobile signal transported over a long distance induces systemic transcriptional gene silencing in a grafted partner. Exp. Bot. 2011, 62, 4561–4570.
Comments 5: [Conclusions: it would be useful to include the more distant aims of the work and its impact on horticulture and agriculture, I suggest rewriting this section.]
Response 5: In conclusions (lines 661-663), we have added more distant aims of our work and its impact on horticulture and agriculture. “These findings can lay a good basis for further epigenetic studies on sweet cherry dwarfing and provide valuable new insight into understanding rootstock-scion interactions.” has been added. Revised portions are marked in red in the paper. Thank you for these positive and constructive comments and suggestions.
Comments 6: [- The main question addressed by the research about Prunus avium inoculation due to DNA methylation and gene expression is presented in the manuscript as a possible correlation.]
Response 6: This study mainly aimed to present a methylome map of the sweet cherry genome, reveal scion widespread DNA methylation alterations caused by dwarfing rootstock, and obtaine abundant genes with methylation and transcription alterations that were potentially involved in rootstock-induced growth changes in the sweet cherry scion. Hence, we plan to conduct validation experiments on these genes in Pd1 dwarfing rootstock leading to dwarfing of sweet cherry scion. Thank you for your positive and constructive comments and suggestions, which provide important guidance for our future research.
Comments 7: [- The topic original or relevant to the field with increasing climate change, we need to pay a lot of attention to the production of food crops and to present other new methods - and this manuscript is a good example.]
Response 7: Thank you very much for the positive comments on our work.
Comments 8: [- It works on a classic topic in the context of genetic modification and DNA methylation. - In the review I wrote: keywords, abstract, discussion, conclusion chapters should be modified. - A deeper discussion is needed between own literature and previous publications.]
Response 8: We have tried our best to revise the manuscript's keywords, abstract, discussion, conclusion chapters. In addition, we have provided a more in-depth discussion based on our own research and previous publications (For detailed responses, please refer to Responses 1-5). Revised portions are marked in red in the paper. Thank you for these positive and constructive comments and suggestions.
We would like to express our great appreciation to you for comments on our paper. Looking forward to hearing from you.
Thank you and best regards.
Yours sincerely,
Corresponding author: Xiaopeng Wen
wenxp@gzu.edu.cn (X.W.)

Round 2
Reviewer 1 Report
Comments and Suggestions for Authors
Regardless of the reason (you couldn't generate quality libraries) and the bibliographies using less than 3 biological replicates (I should point out that very few papers have less than 3 replicates, with the vast majority having at least 3 replicates in omics data and of course, in WGBS, just like you in the RNA-seq data). This paper is based on epigenetic data (thus playing a key role in their analysis), so I believe it should not be accepted for publication with 2 biological replicates.
Author Response
Single-base Methylome Analysis of Sweet Cherry (Prunus avium L.) on Dwarfing Rootstocks Reveals Epigenomic Differences Associated with Scions Dwarfing Conferred by Grafting
(Manuscript ID: ijms-3175946)
Dear Reviewer:
Thanks for your comments concerning our manuscript entitled “Single-base Methylome Analysis of Sweet Cherry (Prunus avium L.) on Dwarfing Rootstocks Reveals Epigenomic Differences Associated with Scions Dwarfing Conferred by Grafting” (Manuscript ID: ijms-3175946). Those comments are all valuable and very helpful for revising and improving our paper, as well as the important guiding significance to our researches. We have studied comments carefully and have made correction which we hope meet with approval. Revised portion are marked in red in the paper. The main corrections in the paper and the responds to the reviewer’s comments are as flowing:
Responds to the reviewer’s comments:
For the Reviewer 1:
Comments: [Regardless of the reason (you couldn't generate quality libraries) and the bibliographies using less than 3 biological replicates (I should point out that very few papers have less than 3 replicates, with the vast majority having at least 3 replicates in omics data and of course, in WGBS, just like you in the RNA-seq data). This paper is based on epigenetic data (thus playing a key role in their analysis), so I believe it should not be accepted for publication with 2 biological replicates.]
Response: Thank you for your rigorous suggestion!
(1)According to the methodology article of WGBS analysis, two biological replicates are feasible. (1.1) According to the “Standards and Guidelines for Whole Genome Shotgun Bisulfite Sequencing” approved by the National Institutes of Health (NIH) Roadmap Epigenomics Mapping Consortium, in order to ensure that the data are reproducible, experiments should be performed with two or more biological replicates. A full DNA methylome should have at least 30X coverage of the genome when reads from biological replicates are combined. For example, a methylome with 2 biological replicates, each with 15X coverage, would be sufficient (Figure R1) [1]. In our experiments, as shown in Table 1 of our paper, 2 biological replicates were used, each with >15X coverage. (1.2) Investigating differential methylation is usually one of the primary goals of doing WGBS, and in the methodology article of methylation analysis by WGBS published in Nature Methods (Figure R2), this article found that decreasing the samples per group from three to two resulted in a modest drop in sensitivity from 77% to 72% at 10X coverage and clearly points out that at least 2 biological replicates are required for DMRs analysis [2]. Hence, two biological replicates in WGBS are feasible. In our paper, two biological replicates were sufficient for data analysis.
(2) In recent reports, although using three biological replicates in WGBS is the mainstream, there are also some articles with two biological replicates that use the above biological replicate setting method (two replicates, each with 15X coverage). In these articles, epigenetic data also plays a key role, and two-replicate data were used, like in orange [3], strawberry [4], pineapple [5], and apple [6], etc. The above articles were published in mainstream scientific journals such as PNAS, Genome Biol., and Plant Biotechnol. J., indicating that experimental designs with 2 biological replicates in WGBS are also peer-reviewed and recognized. In addition, recent articles published in Int. J. Mol. Sci. also include WGBS with fewer than three biological replicates (epigenetic data play an important part in these articles), such as Int. J. Mol. Sci. 2024, 25, 7702 [7]; Int. J. Mol. Sci. 2024, 25, 1118 [8]; Int. J. Mol. Sci. 2023, 24, 3978 [9]; Int. J. Mol. Sci. 2022, 23, 5147 [10]. Therefore, using two biological replicates for WGBS analysis is recognized and accepted.
In the 4.6. DNA Extraction and Whole-Genome Bisulfite Sequencing (lines 591-594), “Referring to the “Standards and Guidelines for Whole Genome Shotgun Bisulfite Sequencing” approved by the National Institutes of Health (NIH) Roadmap Epigenomics Mapping Consortium and the methods reported by Ziller et al., we set up two biological replicates for each group, each replicate with at least 15X coverage [53,54]. ” has been added. Revised portion are marked in red in the paper.
Currently added references:
[53] NIH Roadmap Epigenomics Mapping Consortium. Standards and Guidelines for Whole Genome Shotgun Bisulfite Sequencing. Available online: http://www.roadmapepigenomics.org/files/protocols/data/dna-methylation/MethylC-SeqStandards_FINAL.pdf (accessed on 4 September 2024)
[54] Ziller, M.J., Hansen, K.D., Meissner, A., Aryee, M.J. Coverage recommendations for methylation analysis by whole-genome bisulfite sequencing. Nat. Methods 2015, 12, 230–232.
Thank you very much for your valuable comments, which will provide important guidance for our future research. We will conduct WGBS analysis using three or more biological replicates in our future studies. Thank you again for your kind and help comments on our work.
References
[1] NIH Roadmap Epigenomics Mapping Consortium. Standards and guidelines for whole genome shotgun bisulfite sequencing, Available online: http://www.roadmapepigenomics.org/files/protocols/data/dna-methylation/MethylC-SeqStandards_FINAL.pdf (accessed on 4 September 2024)
[2] Ziller, M.J., Hansen, K.D., Meissner, A., Aryee, M.J. Coverage recommendations for methylation analysis by whole-genome bisulfite sequencing. Nat. Methods 2015, 12, 230–232.
[3] Huang, H.; Liu, R.; Niu, Q.; Tang, K.; Zhang, B.; Zhang, H.; Chen, K.; Zhu, J.K.; Lang, Z. Global increase in DNA methylation during orange fruit development and ripening. PNAS 2019, 22, 116, 1430–1436.
[4] Cheng. J.; Niu, Q.; Zhang, B.; Chen, K.; Yang, R.; Zhu, J.K.; Zhang, Y.; Lang, Z. Downregulation of RdDM during strawberry fruit ripening. Genome Biol. 2018, 19, 212.
[5] Liu, Y.; Priyadarshani, S.V.G.N.; Chi, M.; Yan, M.; Mohammadi, M.A., Zhang, M.; Zhou, Q.; Wang, L.; Luo, T.; Wai, M.H.; Wang, X.; Cai, H.; Wang, H.; Qin, Y. Epigenetic modification mechanisms of chloroplasts mutants in pineapple leaves during somatic regeneration. Hortic. Plant J. 2023, 9, 509–522.
[6] Xu, J.; Zhou, S.; Gong, X.; Song, Y.; van Nocker, S.; Ma, F.; Guan, Q. Single-base methylome analysis reveals dynamic epigenomic differences associated with water deficit in apple. Plant Biotechnol. J. 2018, 16, 672–687.
[7] Tan, B.; Xiao, L.; Wang, Y.; Zhou, C.; Huang, H.; Li, Z.; Hong, L.; Cai, G.; Wu, Z.; Gu, T. Comprehensive Analysis of Placental DNA Methylation Changes and Fetal Birth Weight in Pigs. Int. J. Mol. Sci. 2024, 25, 7702.
[8] Weng, J.; Wang, H.; Cheng, D.; Liu, T.; Zeng, D.; Dai, C.; Luo, C. The Effects of DNA Methylation on Cytoplasmic Male Sterility in Sugar Beet. Int. J. Mol. Sci. 2024, 25, 1118.
[9] Li, Y.; Guo, D. Transcriptome and DNA Methylome Analysis of Two Contrasting Rice Genotypes under Salt Stress during Germination. Int. J. Mol. Sci. 2023, 24, 3978.
[10] Kong, C.; Su, H.; Deng, S.; Ji, J.; Wang, Y.; Zhang, Y.; Yang, L.; Fang, Z.; Lv, H. Global DNA Methylation and mRNA-miRNA Variations Activated by Heat Shock Boost Early Microspore Embryogenesis in Cabbage (Brassica oleracea). Int. J. Mol. Sci. 2022, 23, 5147.
We would like to express our great appreciation to you for comments on our paper. Looking forward to hearing from you.
Thank you and best regards.
Yours sincerely,
Corresponding author: Xiaopeng Wen
wenxp@gzu.edu.cn (X.W.)

Reviewer 2 Report
Comments and Suggestions for Authors
I recommand it for publication
Author Response
Single-base Methylome Analysis of Sweet Cherry (Prunus avium L.) on Dwarfing Rootstocks Reveals Epigenomic Differences Associated with Scions Dwarfing Conferred by Grafting
(Manuscript ID: ijms-3175946)
Dear Reviewer:
Thanks for your comments concerning our manuscript entitled “Single-base Methylome Analysis of Sweet Cherry (Prunus avium L.) on Dwarfing Rootstocks Reveals Epigenomic Differences Associated with Scions Dwarfing Conferred by Grafting” (Manuscript ID: ijms-3175946). Those comments are all valuable and very helpful for revising and improving our paper, as well as the important guiding significance to our researches. We have studied comments carefully and have made correction which we hope meet with approval. Revised portion are marked in red in the paper. The main corrections in the paper and the responds to the reviewer’s comments are as flowing:
Responds to the reviewer’s comments:
For the Reviewer 2:
Comments 1: [I recommand it for publication.]
Response 1: Thank you very much for your positive comments on our work and for recommending the publication of our work.
We would like to express our great appreciation to you for comments on our paper. Looking forward to hearing from you.
Thank you and best regards.
Yours sincerely,
Corresponding author: Xiaopeng Wen
wenxp@gzu.edu.cn (X.W.)

Round 3
Reviewer 1 Report
Comments and Suggestions for Authors
I will insist on my view that at least 3 replicates are needed to draw safe conclusions about the analysis of WGBS, however, unless the editor objects to the analysis (as there is no clear guideline in the journal for three replicates in omic data) may accept the manuscript for publication at IJMS. Personally, I would accept both one and two replications of the WGBS data, as long as these would form part of a decision-making process, e.g., on how we proceed in the functional analysis of genes, thus excluding any candidate genes and/or transcription factors with this analysis. This is also the largest shortcoming of the study, as the authors successfully addressed the rest of my comments.
Author Response
Single-base Methylome Analysis of Sweet Cherry (Prunus avium L.) on Dwarfing Rootstocks Reveals Epigenomic Differences Associated with Scions Dwarfing Conferred by Grafting
(Manuscript ID: ijms-3175946)
Dear Reviewer 1:
Thanks for your comments concerning our manuscript entitled “Single-base Methylome Analysis of Sweet Cherry (Prunus avium L.) on Dwarfing Rootstocks Reveals Epigenomic Differences Associated with Scions Dwarfing Conferred by Grafting” (Manuscript ID: ijms-3175946). Thank you very much for your valuable comments, which will provide important guidance for our future research.
Responds to the reviewer 1’s comments:
Comments: [I will insist on my view that at least 3 replicates are needed to draw safe conclusions about the analysis of WGBS, however, unless the editor objects to the analysis (as there is no clear guideline in the journal for three replicates in omic data) may accept the manuscript for publication at IJMS. Personally, I would accept both one and two replications of the WGBS data, as long as these would form part of a decision-making process, e.g., on how we proceed in the functional analysis of genes, thus excluding any candidate genes and/or transcription factors with this analysis. This is also the largest shortcoming of the study, as the authors successfully addressed the rest of my comments.]
Response: Although you personally prefer three biological replicates! Thank you for recognizing that two biological replicates are feasible in our article.
(1) In the second round submission (revised version), we have given a detailed explanation on the feasibility of two biological replicates (the “Standards and Guidelines for Whole Genome Shotgun Bisulfite Sequencing” approved by the National Institutes of Health (NIH) Roadmap Epigenomics Mapping Consortium, the methodology article of methylation analysis by WGBS published in Nature Methods, as well as in mainstream journals such as PNAS, Genome Biol., and Plant Biotechnol. J., including the recently published IJMS papers, all information proved that two biological replicates are feasible, see Appendix 1 for details).
(2) If additional biological replicate is needed, at present, the sampling season has passed, and there is no same material, so it is temporarily impossible to add a set of biological replicates.
We would like to express our great appreciation to you for comments on our paper. Looking forward to hearing from you.
Thank you and best regards.
Yours sincerely,
Corresponding author: Xiaopeng Wen
wenxp@gzu.edu.cn (X.W.)
Appendix 1
(1)According to the methodology article of WGBS analysis, two biological replicates are feasible. (1.1) According to the “Standards and Guidelines for Whole Genome Shotgun Bisulfite Sequencing” approved by the National Institutes of Health (NIH) Roadmap Epigenomics Mapping Consortium, in order to ensure that the data are reproducible, experiments should be performed with two or more biological replicates. A full DNA methylome should have at least 30X coverage of the genome when reads from biological replicates are combined. For example, a methylome with 2 biological replicates, each with 15X coverage, would be sufficient (Figure R1) [1]. In our experiments, as shown in Table 1 of our paper, 2 biological replicates were used, each with >15X coverage. (1.2) Investigating differential methylation is usually one of the primary goals of doing WGBS, and in the methodology article of methylation analysis by WGBS published in Nature Methods (Figure R2), this article found that decreasing the samples per group from three to two resulted in a modest drop in sensitivity from 77% to 72% at 10X coverage and clearly points out that at least 2 biological replicates are required for DMRs analysis [2]. Hence, two biological replicates in WGBS are feasible. In our paper, two biological replicates were sufficient for data analysis.
(2) In recent reports, although using three biological replicates in WGBS is the mainstream, there are also some articles with two biological replicates that use the above biological replicate setting method (two replicates, each with 15X coverage). In these articles, epigenetic data also plays a key role, and two-replicate data were used, like in orange [3], strawberry [4], pineapple [5], and apple [6], etc. The above articles were published in mainstream scientific journals such as PNAS, Genome Biol., and Plant Biotechnol. J., indicating that experimental designs with 2 biological replicates in WGBS are also peer-reviewed and recognized. In addition, recent articles published in Int. J. Mol. Sci. also include WGBS with fewer than three biological replicates (epigenetic data play an important part in these articles), such as Int. J. Mol. Sci. 2024, 25, 7702 [7]; Int. J. Mol. Sci. 2024, 25, 1118 [8]; Int. J. Mol. Sci. 2023, 24, 3978 [9]; Int. J. Mol. Sci. 2022, 23, 5147 [10]. Therefore, using two biological replicates for WGBS analysis is recognized and accepted.
References
[1] NIH Roadmap Epigenomics Mapping Consortium. Standards and guidelines for whole genome shotgun bisulfite sequencing, Available online: http://www.roadmapepigenomics.org/files/protocols/data/dna-methylation/MethylC-SeqStandards_FINAL.pdf (accessed on 4 September 2024)
[2] Ziller, M.J., Hansen, K.D., Meissner, A., Aryee, M.J. Coverage recommendations for methylation analysis by whole-genome bisulfite sequencing. Nat. Methods 2015, 12, 230–232.
[3] Huang, H.; Liu, R.; Niu, Q.; Tang, K.; Zhang, B.; Zhang, H.; Chen, K.; Zhu, J.K.; Lang, Z. Global increase in DNA methylation during orange fruit development and ripening. PNAS 2019, 22, 116, 1430–1436.
[4] Cheng. J.; Niu, Q.; Zhang, B.; Chen, K.; Yang, R.; Zhu, J.K.; Zhang, Y.; Lang, Z. Downregulation of RdDM during strawberry fruit ripening. Genome Biol. 2018, 19, 212.
[5] Liu, Y.; Priyadarshani, S.V.G.N.; Chi, M.; Yan, M.; Mohammadi, M.A., Zhang, M.; Zhou, Q.; Wang, L.; Luo, T.; Wai, M.H.; Wang, X.; Cai, H.; Wang, H.; Qin, Y. Epigenetic modification mechanisms of chloroplasts mutants in pineapple leaves during somatic regeneration. Hortic. Plant J. 2023, 9, 509–522.
[6] Xu, J.; Zhou, S.; Gong, X.; Song, Y.; van Nocker, S.; Ma, F.; Guan, Q. Single-base methylome analysis reveals dynamic epigenomic differences associated with water deficit in apple. Plant Biotechnol. J. 2018, 16, 672–687.
[7] Tan, B.; Xiao, L.; Wang, Y.; Zhou, C.; Huang, H.; Li, Z.; Hong, L.; Cai, G.; Wu, Z.; Gu, T. Comprehensive Analysis of Placental DNA Methylation Changes and Fetal Birth Weight in Pigs. Int. J. Mol. Sci. 2024, 25, 7702.
[8] Weng, J.; Wang, H.; Cheng, D.; Liu, T.; Zeng, D.; Dai, C.; Luo, C. The Effects of DNA Methylation on Cytoplasmic Male Sterility in Sugar Beet. Int. J. Mol. Sci. 2024, 25, 1118.
[9] Li, Y.; Guo, D. Transcriptome and DNA Methylome Analysis of Two Contrasting Rice Genotypes under Salt Stress during Germination. Int. J. Mol. Sci. 2023, 24, 3978.
[10] Kong, C.; Su, H.; Deng, S.; Ji, J.; Wang, Y.; Zhang, Y.; Yang, L.; Fang, Z.; Lv, H. Global DNA Methylation and mRNA-miRNA Variations Activated by Heat Shock Boost Early Microspore Embryogenesis in Cabbage (Brassica oleracea). Int. J. Mol. Sci. 2022, 23, 5147.
